# Nutrient-dependent regulation of a stable intron modulates germline mitochondrial quality control

Annabel Qi En Ng[1,3], Seow Neng Chan [1,3] & Jun Wei Pek [1,2] ✉

Mitochondria are inherited exclusively from the mothers and are required for the proper development of embryos. Hence, germline mitochondrial quality is highly regulated during oogenesis to ensure oocyte viability. How nutrient availability influences germline mitochondrial quality control is unclear. Here we find that fasting leads to the accumulation of mitochondrial clumps and oogenesis arrest in *Drosophila*. Fasting induces the downregulation of the DIP1-Clueless pathway, leading to an increase in the expression of a stable intronic sequence RNA called *sisR-1*. Mechanistically, *sisR-1* localizes to the mitochondrial clumps to inhibit the poly-ubiquitination of the outer mitochondrial protein Porin/VDAC1, thereby suppressing p62-mediated mitophagy. Alleviation of the fasting-induced high *sisR-1* levels by either *sisR-1* RNAi or refeeding leads to mitophagy, the resumption of oogenesis and an improvement in oocyte quality. Thus, our study provides a possible mechanism by which fasting can improve oocyte quality by modulating the mitochondrial quality control pathway. Of note, we uncover that the *sisR-1* response also regulates mitochondrial clumping and oogenesis during protein deprivation, heat shock and aging, suggesting a broader role for this mechanism in germline mitochondrial quality control.

In animals, mitochondrial inheritance occurs exclusively through the mothers' oocytes. Quality mitochondria not only improve offspring survival and health but are also important for maintaining oocyte competence. In *Drosophila*, the quality and dynamics of the mitochondria are regulated throughout oogenesis and heavily influenced by nutrition and age. During early oogenesis in the germarium, replication of mitochondrial DNA occurs, and mitochondria are fragmented and selected for by a process called mitophagy[1–4]. As oogenesis progresses, mitochondria enter the oocyte by moving along the fusome (an Actin-rich structure that connects the germline cells in a cyst) where they form the Balbiani body[5]. The mitochondria are then trafficked towards the oocyte poleplasm during stage 10 of oogenesis to ensure inheritance of mitochondria by the offspring primordial germ cells[6]. Towards late oogenesis, mitochondria also undergo active

remodeling to enter respiratory quiescence for subsequent storage in the mature oocyte[7,8]. Starvation is known to induce mitochondrial elongation and clumping in the nurse cells, while aging causes mitochondrial fragmentation in germline stem cells (GSCs)[9,10].

Mitochondrial clumping is a general sign of unhealthy mitochondria. Nutrient deprivation, induction of oxidative damage[10,11] and mutations of genes that are involved in mitochondrial related functions, such as *clueless* (*clu*), will result in mitochondrial clumping[12–14], suggesting that accumulation of mitochondrial clumps is a sign of mitochondrial damage.

Stable intronic sequence (sis)RNAs are intron-containing long noncoding (lnc)RNAs that are not rapidly degraded in cells[15,16]. First discovered in *Xenopus tropicalis* oocytes, sisRNAs were later found to be conserved from yeast to human[17–25]. Studies in *Drosophila* have

[1]Temasek Life Sciences Laboratory, 1 Research Link National University of Singapore, Singapore 117604, Singapore. [2]Department of Biological Sciences, National University of Singapore, 14 Science Drive, Singapore 117543, Singapore. [3]These authors contributed equally: Annabel Qi En Ng, Seow Neng Chan. ✉e-mail: junwei@tll.org.sg

revealed that sisRNAs function to regulate various processes during oogenesis and embryonic development[20–22,26–30]. Previous studies have focused on the roles of sisRNAs in regulating gene expression[20,31], however, it is unknown whether sisRNAs exhibit other modes of actions such as regulating organelle dynamics.

It was previously reported that the abundance of several sisRNAs was upregulated in the ovaries during fasting (or starvation) in *Drosophila*[26–28,32]. Upregulation of 4 such sisRNAs (*sisR-1* to *sisR-4*) appeared to inhibit the production of mature oocytes by blocking the oogenesis progression of stage 8/9 egg chambers, which is known to be highly sensitive to nutrient availability. The molecular and cellular mechanisms of how sisRNAs are regulated and how they modulate oogenesis during fasting are not known.

Here we show that *sisR-1* is upregulated and localizes to mitochondrial clumps in *Drosophila* nurse cells during fasting. Knocking down *sisR-1* or refeeding promotes the clearance of these fasting-induced mitochondrial clumps, thereby improving oocyte quality. The abundance of *sisR-1* is regulated by nuclear DIP1 and cytoplasmic Clueless (Clu), and the expression level or localization of both are sensitive to nutrient availability. Mechanistically, *sisR-1* inhibits defective mitochondrial clearance (or mitophagy) by suppressing Parkin (Park)-mediated polyubiquitination of VDAC1/Porin. Thus, our study reveals a nutrient-sensitive pathway consisting of DIP1, Clu and *sisR-1* that modulates mitophagy during oogenesis in *Drosophila*.

## Results

### Fasting promotes *sisR-1*-mediated mitochondrial clumping

We first examined the effects of *sisR-1* knockdown (hereafter *sisR-1* RNAi) on the production of stage 14 oocytes (or mature oocytes) under fed and fasted conditions (Fig. 1A). Newly eclosed female flies were fed or fasted for 2 days before the ovaries were dissected to count the number of stage 14 oocytes present (see Methods). Under fed conditions, the stage 14 oocyte count was lower in *sisR-1* RNAi flies (2 independent RNAi lines) than the controls, a phenotype that is likely explained by a reduction in GSCs in *sisR-1* RNAi flies as reported previously (Fig. 1B)[29]. Consistent with what was observed previously[28], we also observed an increase in the number of stage 14 oocytes produced under fasted conditions in *sisR-1* RNAi flies (Fig. 1B), suggesting that *sisR-1* inhibits the production of mature oocytes during fasting. We further verified the expression of *sisR-1* in the different genotypes under both fed and fasted conditions by performing single molecule fluorescent in situ hybridization (smFISH) (Supplementary Fig. S1).

Also consistent with previous report[28], *sisR-1* was found to be upregulated in the ovaries of fasted and TOR RNAi flies (Fig. 1C,D). To understand the cellular mechanism by which *sisR-1* regulates oogenesis during fasting, we examined the localization of *sisR-1* by performing smFISH. Under fed conditions, *sisR-1* levels remained low and localized predominantly in the nuclei of nurse cells (Fig. 1E). During fasting, an accumulation of *sisR-1* and appearance of mitochondrial clumps could be seen in the nurse cell cytoplasm (Fig. 1E). These mitochondrial clumps were reminiscent of those reported in previous studies[10–14] and localized perinuclearly with width ranging from ~2 μm to ~8 μm (Supplementary Fig. S2). Interestingly, *sisR-1* transcripts were often seen localizing near and inside the mitochondrial clumps (Fig. 1E, F, arrowheads). To confirm the association of *sisR-1* with mitochondria, we performed mitochondrial fractionation and found an increase in the abundance of *sisR-1* transcripts in the mitochondria after fasting (Fig. 1G).

Because mitochondrial clumping is often an indication of unhealthy mitochondria[10–14], we asked if *sisR-1* is required for the production of these mitochondrial clumps during fasting. Indeed, in *sisR-1* RNAi flies, the percentages of stage 8/9 egg chambers containing mitochondrial clumps were reduced (Fig. 1H,I, arrowheads). Furthermore, when we examined egg chambers that still contained some

clumping, the overall number of clumps per egg chamber was also drastically reduced (Fig. 1J). Next, we asked if overexpression of *sisR-1* alone is sufficient to induce clumping of mitochondria without fasting. In support of this idea, we observed that *vasa-Gal4* driven over-expression of *sisR-1* led to the formation of mitochondrial clumps in germline cells (Supplementary Fig. S3). Thus, the upregulation of *sisR-1* is at least partially responsible for the formation of mitochondrial clumps during fasting (Fig. 1K).

### Clueless represses *sisR-1* abundance in the cytoplasm

Since *sisR-1* expression is upregulated during fasting (Fig. 1C)[28], we moved to investigate the mechanism that represses *sisR-1* expression under fed conditions. During fasting, *sisR-1* localizes to mitochondria and promotes clumping. We explored the possibility of a nutrient-sensitive mitochondria-associated RNA-binding protein as a regulator of *sisR-1*. *Drosophila* Clu is a conserved RNA-binding protein that localizes to the surface of mitochondria as discrete cytoplasmic bodies in nurse cells[14,33]. Under nutrient deprivation, Clu bodies disassemble, concomitant with the formation of mitochondrial clumps[34]. Similar clumps were also observed in fed *clu* mutants, suggesting that *clu* suppresses mitochondrial clumping[12,14].

We hypothesized that during fasting, Clu bodies disassemble, leading to the accumulation of mitochondrial *sisR-1*, which causes clumping. In *clu* mutants, *sisR-1* expression was upregulated as measured by RT-qPCR (Supplementary Fig. S4A). *clu* mutant egg chambers also exhibited cytoplasmic *sisR-1* that associates with mitochondrial clumps (Supplementary Fig. S4B, arrowheads), which is a phenotype also seen in fasted wildtype ovaries (Fig. 1E).

Previous studies showed that *clu* is important to facilitate translation of mitochondrial proteins[12]. To circumvent this issue, we observed *clu* heterozygous mutants instead, which had normal translation[12]. Heterozygous *clu* mutants also showed an upregulation of *sisR-1* expression and the colocalization of *sisR-1* with mitochondrial clumps (Fig. 2A,B). To test whether *sisR-1* mediates mitochondrial clumping in *clu* heterozygous mutants, we knocked down *sisR-1* to see if it could rescue the clumping. Consistent with the model, we saw a reduction in the percentages of egg chambers containing mitochondrial clumps as well as the overall number of clumps (Fig. 2C-E).

We next investigated if Clu directly regulates the stability of *sisR-1*. First, by performing immunoprecipitation of endogenous GFP tagged Clu using a protein trap line, we found physical interaction between Clu and *sisR-1* in the ovaries (Fig. 2F). As a negative control, Clu did not interact with U85 RNA (Fig. 2F). Next, alpha-amanitin assay showed that *sisR-1* was more stable in *clu* mutant ovaries compared to controls (Fig. 2G). On the other hand, the stability of a positive control *Arglu1* pre-mRNA was similar in both cases (Supplementary Fig. S4C). Taken together, our data suggest that Clu negatively regulates the levels of *sisR-1* by promoting its decay.

### DIP1 represses *sisR-1* abundance in the nucleus

DIP1 is a nuclear RNA binding protein that binds and promotes *sisR-1* degradation in *Drosophila*[29,35]. To investigate if DIP1 is also involved in the regulation of *sisR-1* during fasting, we first examined the expression of DIP1 in ovaries of fed and fasted flies. RT-qPCR and western blotting revealed that although the levels of *DIP1* mRNA were similar, DIP1 protein abundance was downregulated in ovaries of fasted flies (Fig. 3A,B). Intriguingly, fasting also abolished the nuclear localization of DIP1 protein in those nurse cells (Fig. 3C), suggesting that the downregulation of DIP1 may lead to upregulation of *sisR-1* during fasting.

When we looked at the localizations of *sisR-1* and mitochondria in *DIP1* mutants, we did not observe any obvious differences when compared to *Oregon R* controls (Fig. 3D, Supplementary Fig. S5). The only noticeable difference was the increase in nuclear *sisR-1*, which is

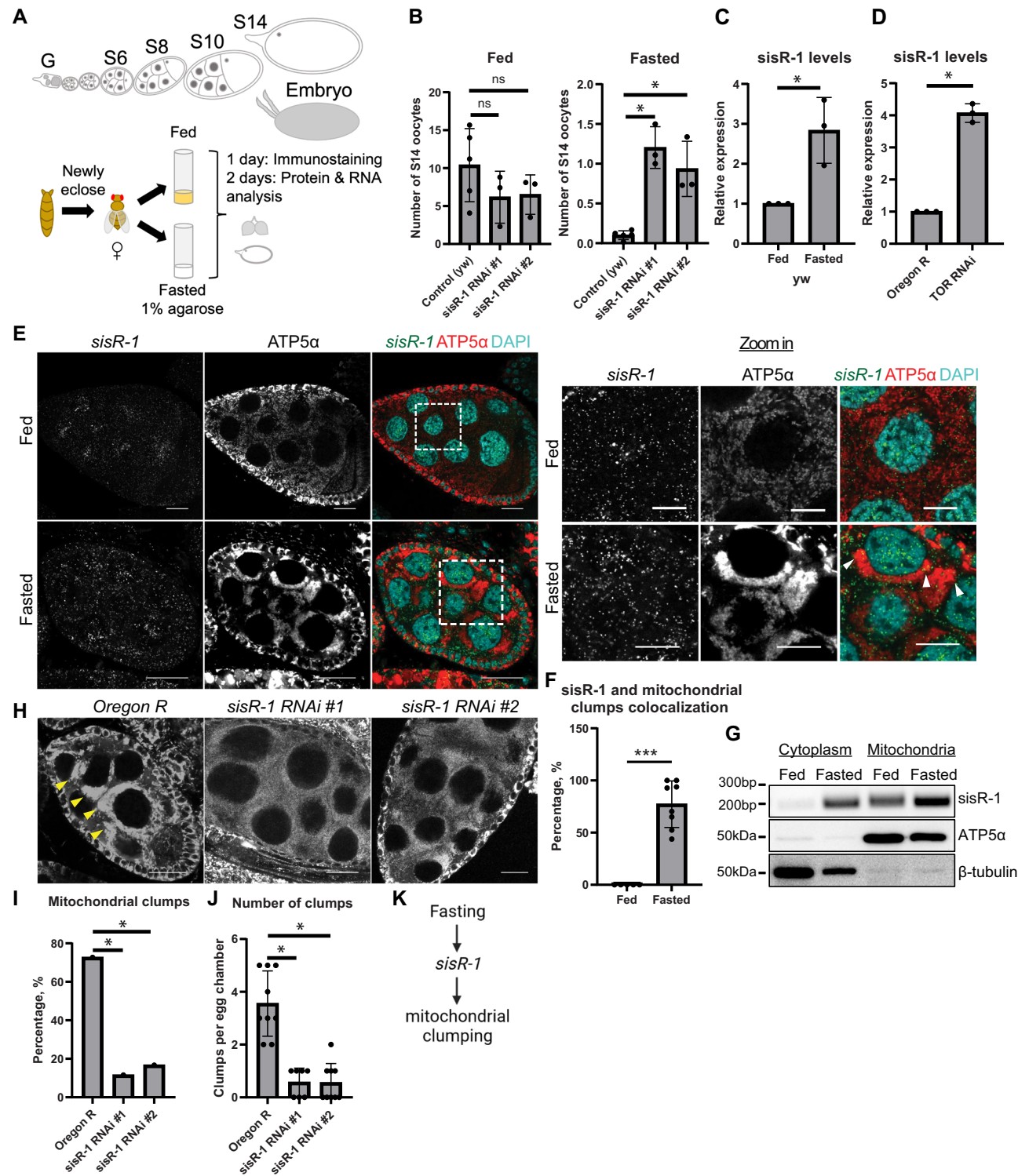

consistent with DIP1 repressing *sisR-1* in the nucleus as reported previously (Fig. 3D, F, Supplementary Fig. S5)[29]. We hypothesized that the lack of cytoplasmic upregulation of *sisR-1* in *DIP1* mutants could be due to functional Clu particles that promote the degradation of *sisR-1* in the cytoplasm. In support of this idea, we observed a more severe upregulation of *sisR-1* by smFISH and RT-qPCR in *DIP1;clu* double mutants, which was concomitant with clumping of mitochondria in the nurse cells (Fig. 3D–G, Supplementary Fig. S5). The size of these ovaries was also significantly smaller compared to single mutants (Fig. 3H), indicating that DIP1 and Clu act synergically in repressing *sisR-1*. During fasting, both the downregulation of nuclear DIP1 and disassembly of

cytoplasmic Clu bodies lead to the upregulation of *sisR-1* in the cytoplasm, which consequently causes the formation of mitochondrial clumps (Fig. 3I).

## *sisR-1* inhibits p62-mediated clearance of mitochondria during fasting

To investigate the downstream function of *sisR-1* in regulating mitochondria during fasting, we examined p62 (SQSTM1 or *Drosophila* Ref(2)P), which is a receptor of autophagy[36,37]. p62 recognizes ubiquitinated cargos and targets them to autophagosomes for degradation[36]. During mitophagy, which is a specialized pathway of

**Fig. 1** | *sisR-1 regulates mitochondrial clumping during fasting.* **A** Germline cells at different stages of oogenesis. G: germarium, S: stage. **B** Number of stage 14 oocytes per ovary in fed and fasted *y w, sisR-1 RNAi line 1* and *2*. Data are presented as mean values +/−SD. *$p < 0.001$. $n = 30$ ovaries. Two-tailed t-test. **C** Relative levels of *sisR-1* normalized against *actin5C* in *y w* fed versus fasted oocytes. Data are presented as mean values +/−SD from three biological replicates. *$p = 0.01$. Two-tailed t-test. **D** Relative levels of *sisR-1* normalized against *actin5C* in control sibling versus *nos-Gal4 > TOR RNAi* ovaries. Data are presented as mean values +/−SD from $n = 3$ biological replicates. *$p < 0.01$. Two-tailed t-test. **E** Confocal images of stage 8/9 egg chambers showing localization of *sisR-1* (Green), ATP5α (Red) and DAPI (Blue) in *Oregon R* fed with yeast paste for 2 days and fasted for 1 day post eclosure. Boxed region is magnified. Arrowheads point to mitochondrial clumps that colocalize with *sisR-1* in fasted females. Scale bar = 20 μm. **F** Percentages of cytoplasmic *sisR-1* dots

that colocalize with mitochondrial clumps. Data are presented as mean values +/−SD. ***$p < 0.001$. $n = 5$ (fed), 8 (fasted) slices. Two-tailed t-test. **G** Mitochondria isolation showing enrichment of *sisR-1* in mitochondrial fractions. ATP5α and β-Tubulin were used as positive controls. $N = 2$ biological replicates. **H** Confocal images of stage 8/9 egg chambers showing ATP5α localization in fasted *Oregon R, sisR-1 RNAi line 1 and 2*. Arrowheads point to mitochondrial clumps in fasted *Oregon R*. Scale bar = 20 μm. **I** Percentages of total stage 8/9 egg chambers with mitochondrial clumps in *Oregon R* and *sisR-1 RNAi line 1 and 2* ($n = 100$ egg chambers) *$p < 0.001$. Chi-square test, one-sided. **J** Number of clumps per egg chamber present in stage 8/9 egg chamber that contain clumps. Data are presented as mean values +/−SD. *$p < 0.01$. $n = 9$ (*Oregon R*), 7 (RNAi #1), 9 (RNAi #2) egg chambers. Two-tailed t-test. **K** Proposed model on how fasting affects *sisR-1* levels and causes mitochondrial clumping. Source data are provided as a Source Data file.

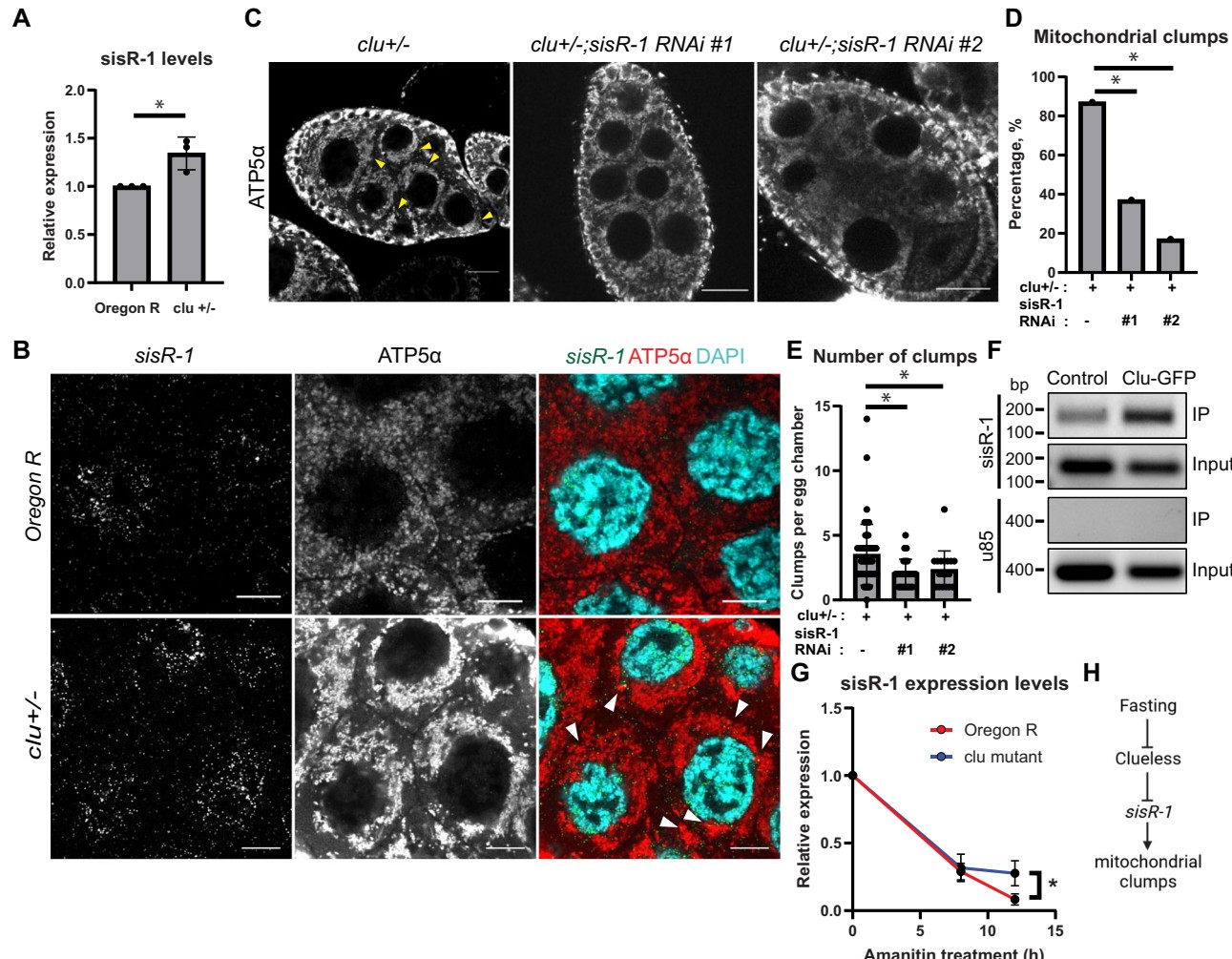

**Fig. 2** | *clueless regulates sisR-1 expression.* **A** Relative expression of *sisR-1* in *Oregon R* and *clu[d08713]* heterozygous stage 14 oocytes. Data are presented as mean values +/−SD from three biological replicates. *$p < 0.05$. Two-tailed t-test. **B** Confocal images of stage 8/9 egg chambers showing localization of *sisR-1* (Green), ATP5α (Red) and DAPI (Blue) in fed *Oregon R* and *clu[d08713]* heterozygous. Arrowheads point to colocalization of *sisR-1* and mitochondrial clumps. Scale bar = 10 μm. **C** Confocal images of stage 8/9 egg chambers showing localization of ATP5α in *clu[d08713]/+, clu[d08713]/+;sisR-1 RNAi line 1 and 2*. Arrowheads point to mitochondrial clumps. Scale bar = 20 μm. **D** Percentages of stage 8/9 egg chambers with mitochondrial clumps in *clu[d08713]/+, clu[d08713]/+;sisR-1 RNAi*

*line 1 and 2* ($n = 100$ egg chambers) *$p < 0.001$. Chi-square test, one-sided. **E** Number of mitochondrial clumps present in stage 8/9 egg chambers that contain clumps in *clu[d08713]/+, clu[d08713]/+;sisR-1 RNAi line 1 and 2*. Data are presented as mean values +/−SD. **$p < 0.05$. $n = 54$ (*clu/+*), 25 (*clu/+;RNAi #1*), 18 (*clu/+;RNAi #2*) egg chambers. Two-tailed t-test. **F** Enrichment of *sisR-1*, but not *U8S*, in Clu-GFP immunoprecipitate compared to IgG control. **G** Relative levels of *sisR-1* normalized to *Rp49* after alpha-amanitin treatment in *Oregon R* and *clu* ovaries. Data are presented as mean values +/−SD from three biological replicates. *$p < 0.05$. Two-tailed t-test. **H** Model on the effects of fasting on Clu protein and *sisR-1* in promoting mitochondrial clump formation. Source data are provided as a Source Data file.

autophagy that targets mitochondria, p62 binds to ubiquitinated outer mitochondrial membrane proteins and selectively targets them for degradation (Fig. 4A)[38]. During this process, p62 is similarly degraded simultaneously.

When fed, p62 forms small cytoplasmic bodies in the nurse cell cytoplasm (Supplementary Fig. S6A). We examined the localization of p62 in relation to *sisR-1* and mitochondrial clumps in ovaries of fasted flies (Fig. 4B, C). Unexpectedly, we found that majority (~70%) of the

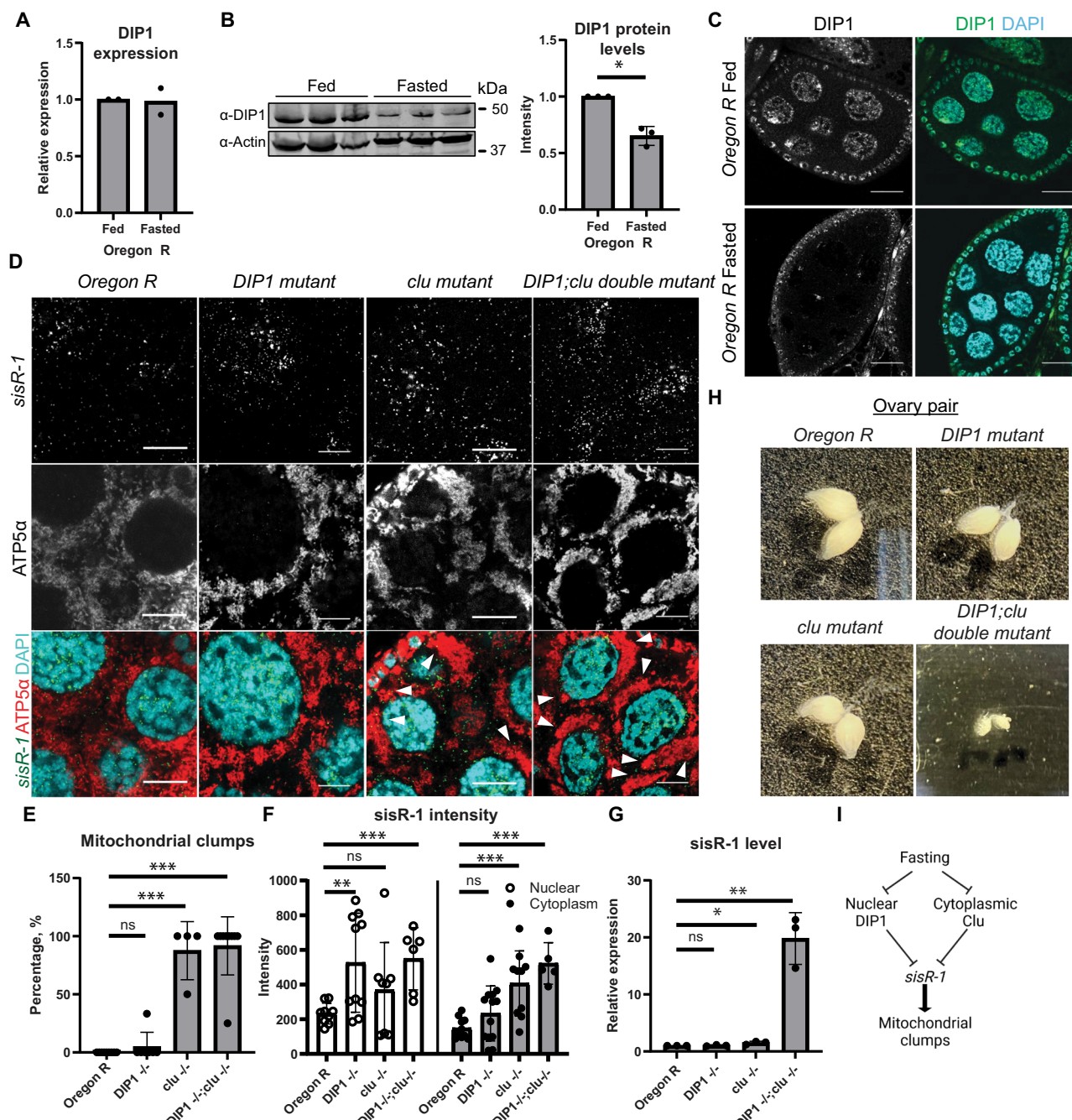

**Fig. 3 | DIP1 regulates *sisR-1* expression during fasting. A** Relative expression of *DIP1* normalized to *actin5C* in fed and fasted *Oregon R* stage 14 oocytes. Error bars depict SD from two biological replicates. ns: not significant, *p* > 0.05. Two-tailed t-test. **B** Levels of DIP1 protein in ovaries of fed and fasted *Oregon R* flies. Graph shows the quantified relative levels of DIP1 normalized to Actin. *p* < 0.05. Data are presented as mean values +/−SD from three biological replicates. Two-tailed t-test. **C** Confocal images of stage 8/9 egg chambers showing localization of DIP1 (Green) and DAPI (Blue) in fed and fasted *Oregon R*. Scale bar = 20 μm. **D** Confocal images of stage 8/9 egg chambers showing localization of *sisR-1* (Green), ATP5α (Red) and DAPI (Blue) in fed *Oregon R*, *DIP1−/−*, *clu−/−* and *DIP1−/−;clu−/−*. Scale bar = 10 μm. **E** Percentages of nurse cell per stage 8/9 egg chamber with mitochondrial clumps in fed *Oregon R*, *DIP1−/−*, *clu−/−* and *DIP1−/−;clu−/−*. Data are presented as mean values +/−SD. ***p* < 0.001. *n* = 9 (*Oregon R*), 7 (*DIP1−/−*), 4 (*clu−/−*) and 9 (*DIP1−/−;clu −/−*) egg chambers counted. Two-tailed t-test. **F** Relative expression of *sisR-1* by smFISH in the nucleus and cytoplasm of stage 8/9 egg chambers in fed *Oregon R*, *DIP1 −/−*, *clu−/−* and *DIP1−/−;clu −/−*. Data are presented as mean values +/−SD. ***p* < 0.001. *n* = 9 (N), 10 (C) for *Oregon R*, 10 (N) and 13 (C) for *DIP1−/−*, 8 (N) and 10 (C) for *clu−/−* and 6 (N) and 5 (C) for *DIP1−/−;clu−/−*. egg chambers. Two-tailed t-test. **G** Relative levels of *sisR-1* in fed *Oregon R*, *DIP1−/−*, *clu−/−* and *DIP1−/−;clu−/−* oocytes. Data are presented as mean values +/−SD from three biological replicates. **p* < 0.05 ***p* < 0.01. Two-tailed t-test. **H** Images of dissected ovary pairs of fed *Oregon R*, *DIP1−/−*, *clu−/−* and *DIP1−/−;clu−/−* taken at 1.5X zoom. **I** Proposed model on the effects of fasting on DIP1, Clu and *sisR-1* in promoting mitochondrial clump formation. Source data are provided as a Source Data file.

aggregates within mitochondrial clumps colocalized with only *sisR-1*, while only ~20% colocalized with only p62. The remaining minority fraction (~10%) colocalized with both *sisR-1* and p62. The high mutual exclusivity of mitochondrial clumps colocalizing with either *sisR-1* or

p62 suggests the possibility that *sisR-1* may suppress active localization of p62 to mitochondrial clumps.

To test if *sisR-1* is involved in the regulation of p62-mediated mitophagy (or autophagy) during fasting, we monitored the levels of

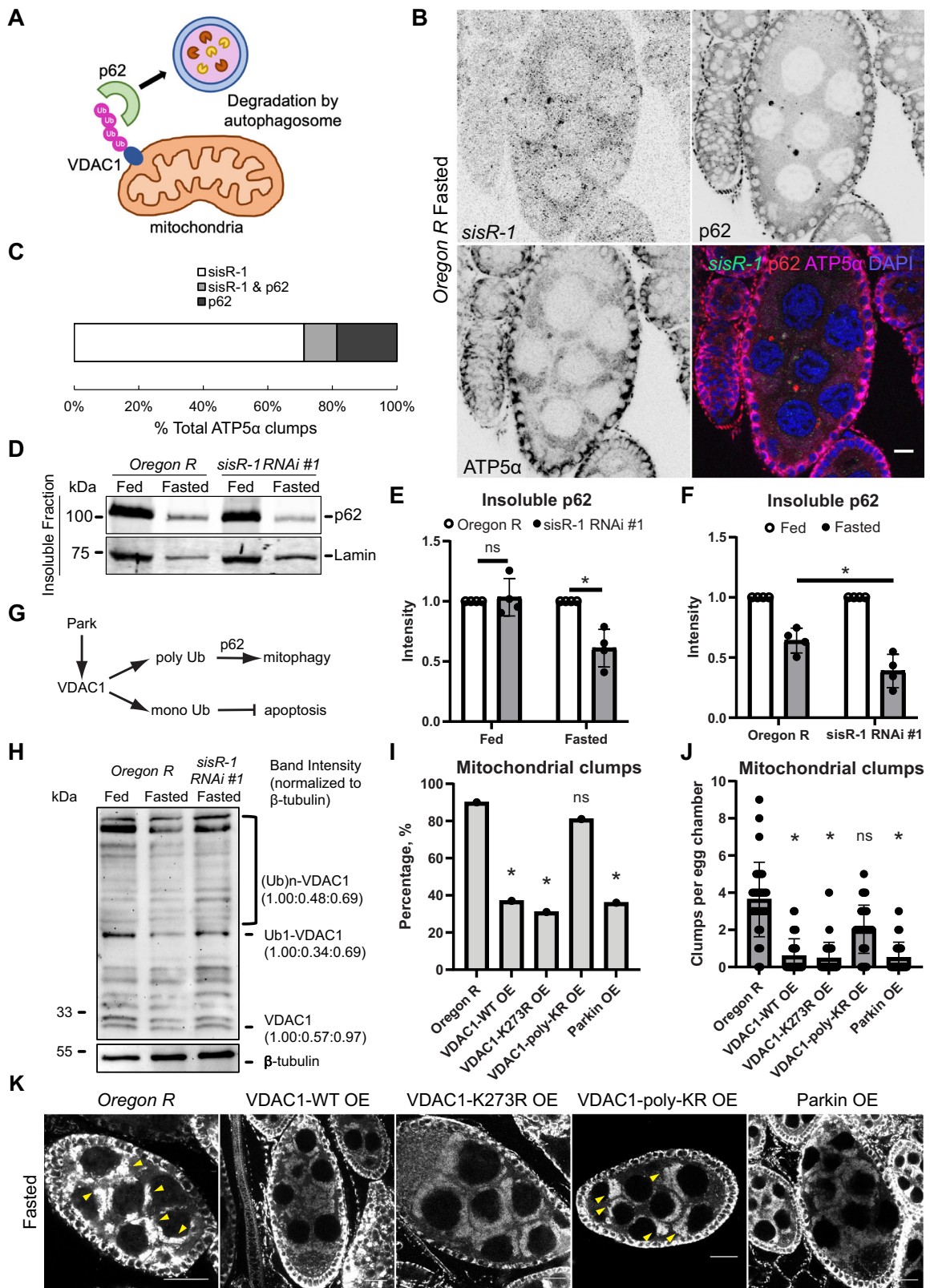

insoluble p62 in the ovaries when flies were fed or fasted[39]. The levels of insoluble p62 were significantly lower in *sisR-1* RNAi ovaries when compared to *Oregon R* controls only under fasting conditions (Fig. 4D, E). To corroborate this finding, we observed that the number of p62 bodies was also lower in *sisR-1* RNAi ovaries under fasting conditions (Supplementary Fig. S6B). These observations suggest that *sisR-1* regulates p62 only during fasting. As expected, the levels of insoluble p62

decreased during fasting in *Oregon R* controls (Fig. 4D, F), which reflected the clearance of p62 via autophagy. Importantly, the degree of p62 downregulation was significantly greater in *sisR-1* RNAi flies (Fig. 4F), supporting the idea that *sisR-1* inhibits p62-mediated autophagy.

It was recently reported in mammalian cells of a vault noncoding RNA that inhibits autophagy by directly binding to p62 to inhibit its oligomerization[40]. To check if *sisR-1* interacts with p62, we performed

**Fig. 4 | *sisR-1* regulates p62-mediated mitophagy via polyubiquitination of VDAC1. A** A drawing of p62-mediated mitophagy. **B** Confocal images showing localizations of *sisR-1* (Green), p62 (Red), ATP5α (Magenta) and DAPI (Blue) in fasted *Oregon R*. *n* = 3 biological replicates. Scale bar = 10 μm. **C** Stacked bar graph showing the percentages of ATP5α-positive mitochondrial clumps that colocalize with *sisR-1*, p62 or both *sisR-1* and p62. **D** Western blots showing levels of p62 in insoluble fractions extracted from ovaries of fed and fasted *Oregon R* and *sisR-1 line 1* flies. **E**, **F** Graphs showing the relative levels of insoluble p62 in fed and fasted *Oregon R* and *sisR-1 RNAi line 1*, as shown in the blot in **D**. Data are presented as mean values +/−SD from four biological replicates. *\*p < 0.05*. Two-tailed t-test. **G** Model on the effects of Park on ubiquitination of VDAC1 affecting p62-mediated mitophagy of defective mitochondria. **H** Western blots showing mono- and poly-ubiquitination of VDAC1 in fed and fasted *Oregon R* and fasted *sisR-1 RNAi line 1* against control β-Tubulin. *n* = 2 biological replicates. **I** Chart showing the

percentages of stage 8/9 egg chambers with mitochondrial clumps in fasted *Oregon R*, *VDAC1-WT* overexpression, *VDAC1-K273R* overexpression, *VDAC1-poly-KR* overexpression and *Parkin* overexpression ovaries. *n* = 30 (*Oregon R*), 37 (WT), 35 (K273R), 30 (poly-KR), 24 (Park OE). egg chambers counted. *\*p < 0.05*. ns not significant. Chi-square test, one-sided. **J** Chart showing the average numbers of mitochondrial clumps present per stage 8/9 egg chamber that contain clumps in fasted *Oregon R*, *VDAC1-WT* overexpression, *VDAC1-K273R* overexpression, *VDAC1-poly-KR* overexpression and *Parkin* overexpression ovaries. Data are presented as mean values +/−SD. *\*p < 0.001*. *n* = 30 (*Oregon R*), 37 (WT), 35 (K273R), 30 (poly-KR), 24 (Park OE). egg chambers counted. Two-tailed t-test. **K** Confocal images of ATP5α localization in fasted *Oregon R*, *VDAC1-WT* overexpression, *VDAC1-K273R* overexpression, *VDAC1-poly-KR* overexpression and *Parkin* overexpression ovaries Arrowheads point to mitochondrial clumps. *n* = 3 biological replicates. Scale bar = 20 μm. Source data are provided as a Source Data file.

immunoprecipitation of GFP-p62 from a transgenic line expressing GFP-p62. GFP-p62 colocalized with endogenous p62 and exhibited similar dynamics during fasting (Supplementary Fig. S6C). Immunoprecipitation of GFP-p62 also pulled down endogenous p62 but not *sisR-1* in both control and *sisR-1* overexpression conditions (Supplementary Fig. S6D, E). Our data suggest that *sisR-1* does not function to regulate p62 through direct binding.

### *sisR-1* inhibits poly-ubiquitination of VDAC1 during fasting

To identify the downstream target of *sisR-1* in the regulation of mitophagy, we considered VDAC1 (or *Drosophila* Porin), which is an outer mitochondrial protein that has been shown to be a p62 target during mitophagy[36]. It was recently shown that the ubiquitination levels of VDAC1 can influence the cell decision to undergo apoptosis or mitophagy[41]. Mono-ubiquitination of VDAC1 inhibits apoptosis, whereas poly-ubiquitination promotes mitophagy via p62 (Fig. 4G). Western blotting of ovarian lysates showed that both mono- and poly-ubiquitination of VDAC1 were downregulated during fasting (Fig. 4H). Furthermore, this downregulation of VDAC1 ubiquitination was dependent on *sisR-1* (Fig. 4H). These results are consistent with a role for *sisR-1* in the downregulation of mitophagy during fasting.

Our results so far suggest that during fasting, upregulation of mitochondrial *sisR-1* inhibits VDAC1 poly-ubiquitination and mitophagy, consequently leading to mitochondrial clumping. Since *Drosophila* Parkin (Park) facilitates mitophagy by promoting the poly-ubiquitination of VDAC1[41], we tested if germline overexpression of Park can suppress the mitochondrial clumping phenotypes during fasting. Indeed, overexpression of Park significantly reduced the number of mitochondrial clumps and the percentages of egg chambers containing clumps (Fig. 4I–K). Consistent with the model, germline overexpression of WT and K273R (mono-ubiquitination mutant), but not poly-KR (poly-ubiquitination mutant), forms of VDAC1 significantly suppressed the mitochondrial clumping phenotypes (Fig. 4I–K). Taken together, our experiments support the model that poly-ubiquitination of VDAC1 is required to promote mitophagy in the ovaries during fasting.

### Clearance of fasting-induced mitochondrial clumps improves oocyte quality

Our results so far show that fasting leads to the upregulation of *sisR-1*, which inhibits the clearance of mitochondrial clumps by mitophagy. Next, we asked if the accumulation of these clumps during fasting has any impact on oocyte quality. First, we compared the hatch rates of fasted *Oregon R* and *sisR-1* RNAi flies and observed a significant improvement from ~45% (*Oregon R*) to ~65% in *sisR-1* RNAi flies, consistent with the model that a reduction of *sisR-1*-mediated mitochondrial clumps improves oocyte quality during fasting (Figs. 1H–J and 5A, B).

Since the expression of *sisR-1* is dynamically modulated by food availability, we tested if clearance of mitochondrial clumps after fasting

by refeeding had any impact on oocyte quality (Fig. 5C). Newly-eclosed *Oregon R* female flies were exposed to either fed or fasted conditions for 30 h before they were transferred to a nutrient-rich environment with males for 24 h. The flies were subsequently removed to determine the hatch rates (Fig. 5C). Interestingly, an initial fasting of 30 h significantly improved the hatch rates as compared to those that are fed for the same duration (Fig. 5D). Furthermore, the effect was only seen when flies were exposed to a longer duration of fasting (for 30 h), but not at 16 h (Fig. 5D). Different feeding durations did not alter the hatch rates (Fig. 5D), indicating that the effects were specific to fasting.

The impact of fasting duration on oocyte quality was further corroborated by the abundance of *sisR-1* and mitochondrial clumps under these conditions. Higher amounts of *sisR-1* and mitochondrial clumps were present in nurse cells fasted for 30 h than 16 h (Fig. 5E–I). However, no significant difference was observed under fed conditions (Fig. 5E–I). These observations suggest that fasting induces the accumulation of *sisR-1*, which in turn leads to mitochondrial clumping, and further clearance of these mitochondrial clumps by either genetic knockdown of *sisR-1* or refeeding led to the production of better-quality oocytes.

### *sisR-1* response in other types of stresses

We also explored other forms of stresses to examine if they alter *sisR-1* levels and mitochondrial clumping. Interestingly, protein deprivation alone was sufficient to induce *sisR-1* expression and mitochondrial clumping (Fig. 6A–D). We also found that subjecting flies to a transient 1-h heat shock at 37 °C also led to an induction of *sisR-1* and mitochondrial clumps (Fig. 6A–D). As a result, we further investigated if this form of transient heat stress could also improve oocyte quality, similar to what we observed with fasting. Indeed, flies subjected to a 1-h heat shock at 37 °C showed improved hatch rates as compared to those kept at 25 °C continuously (Fig. 6E).

We next examined if *sisR-1* and mitochondrial clumps had any impact on oogenesis during aging. Ovaries of young females (4-day old) displayed lower expression of *sisR-1* and less mitochondrial clumps than that of older 14-day old flies (Fig. 6A–D). This was concomitant with a significant decrease in egg production from ~70 eggs/female/day in 4-day old flies to ~43 eggs/female/day in 14-day old flies (Fig. 6F). To investigate if the decrease in egg production during aging was caused by *sisR-1* and mitochondrial clumping, we examined the number of eggs produced by *sisR-1* RNAi flies. In *sisR-1* RNAi flies, the decrease in egg production from day 4 to day 14 was not significant (Fig. 6F). The number of mitochondrial clumps was also significantly lower in 14-day old *sisR-1* RNAi flies compared to control flies of the same age (Fig. 6A, D). Although we observed a decrease in egg production in *Oregon R* during aging, we did not observed any significant differences in hatch rates (Fig. 6F, G). On the other hand, in *sisR-1* RNAi flies, a decrease in hatch rate but not egg production, was observed during aging (Fig. 6F, G). These results suggest that during aging, an increase in *sisR-1* expression may

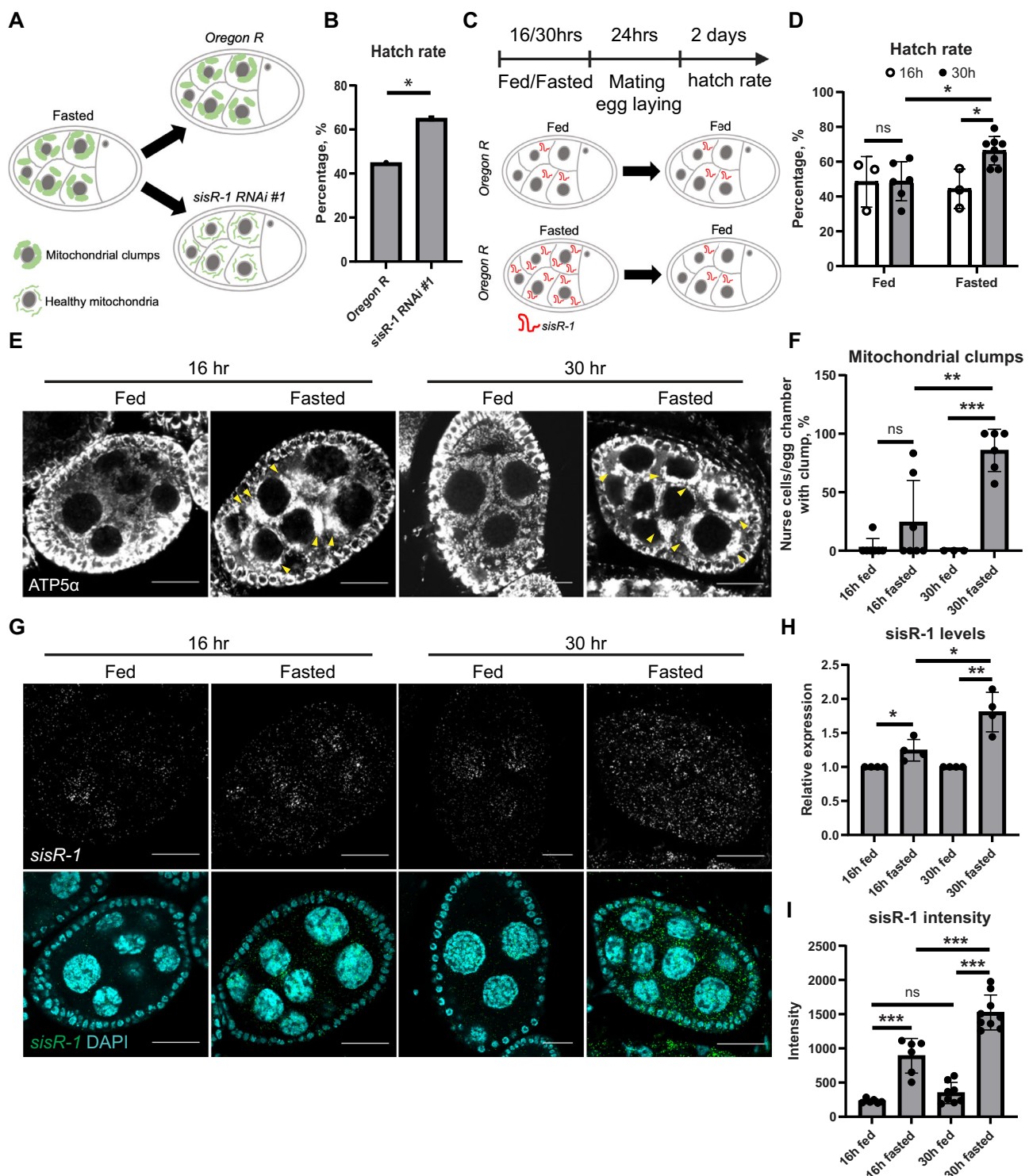

**Fig. 5 | Clearance of fasting-induced mitochondrial clumps improves oocyte quality. A** Stage 8/9 egg chambers showing fasting-induced mitochondrial clumping persists in *Oregon R* but not in *sisR-1 RNAi* ovaries. **B** Hatch rates 2 days after egg laying in fasted *Oregon R* and *sisR-1 RNAi*. *$p < 0.05$. Chi-square test, one-sided. $n = 183$ eggs in 20 experiments (*Oregon R*) and 86 eggs in 7 experiments (RNAi). **C** Stage 8/9 egg chambers showing *sisR-1* expression in initially fed and fasted *Oregon R* flies and after being subsequently fed. **D** Hatch rates of *Oregon R* females that underwent the indicated duration of feeding and fasting regimes. Data are presented as mean values +/−SD. *$p < 0.05$. ns not significant. Two-tailed t-test. $n = 3$ (16 h fed), 3 (16 h fasted), 6 (30 h fed), 8 (30 h fasted) biological replicates. **E** Confocal images of ATP5α localization in fed or fasted *Oregon R* ovaries at 16 or 30 h. Arrowheads point to mitochondrial clumps. Scale bar = 20 μm. **F** Percentages

of nurse cell per stage 8/9 egg chamber with mitochondrial clumps in fed or fasted *Oregon R* ovaries at 16 or 30 h. Data are presented as mean values +/−SD. **$p < 0.01$, ***$p < 0.001$. $n = 7$ (16 h fed), 7 (16 h fasted), 3 (30 h fed), 6 (30 h fasted) egg chambers counted. Two-tailed t-test. **G** Confocal images of *sisR-1* (Green) and DAPI (Blue) localizations in fed or fasted *Oregon R* ovaries at 16 or 30 h. Arrowheads point to mitochondrial clumps. Scale bar = 20 μm. **H** Levels of *sisR-1* normalized to *actinSC* in fed or fasted *Oregon R* ovaries at 16 or 30 h. Data are presented as mean values +/−SD. $n = 4$ biological replicates. Two-tailed t-test. **I** Relative expression of *sisR-1* by smFISH in the cytoplasm of stage 8/9 egg chambers in fed or fasted *Oregon R* ovaries at 16 or 30 h. Data are presented as mean values +/−SD. ***$p < 0.001$. $n = 6$ (16 h fed), 6 (16 h fasted), 8 (30 h fed), 9 (30 h fasted) egg chambers counted. Two-tailed t-test. Source data are provided as a Source Data file.

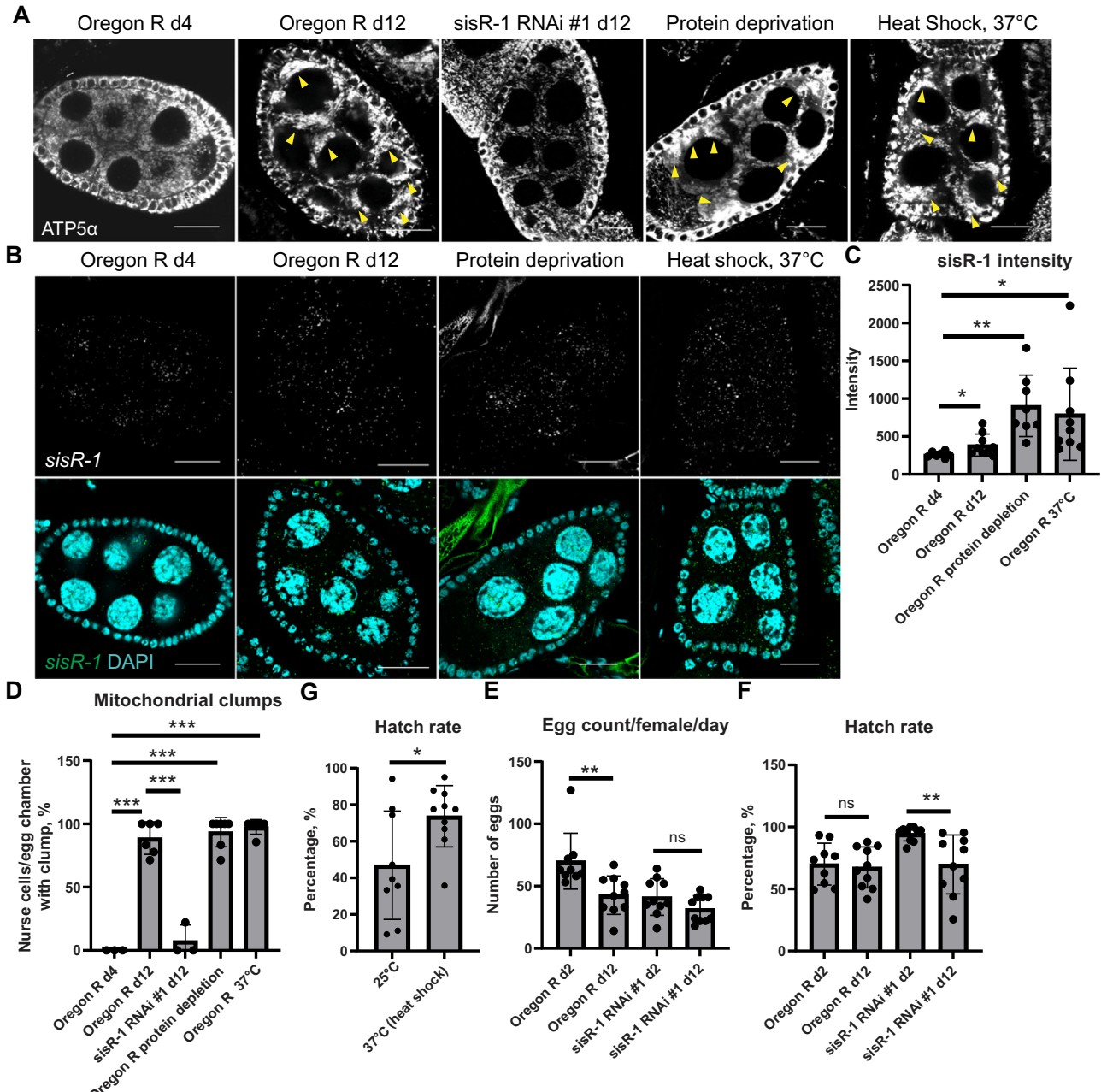

**Fig. 6 | *sisR-1* response in other types of stresses. A** Confocal images of ATP5α localization in ovaries during aging, protein deprivation and after heat shock. Arrowheads point to mitochondrial clumps. Scale bar = 20 μm. **B** Confocal images of *sisR-1* (Green) and DAPI (Blue) localizations in ovaries during aging, protein deprivation and after heat shock. Scale bar = 20 μm. **C** Relative expression of *sisR-1* as measured by smFISH in the cytoplasm of stage 8/9 egg chambers during aging, protein deprivation and after heat shock. Data are presented as mean values +/−SD. *$p < 0.05$, **$p < 0.001$. $n = 7$ (day 4), 9 (day 12), 7 (fed), 8 (protein deprived), 7 (25 °C), 9 (37 °C 1 h) egg chambers counted. Two-tailed t-test. **D** Chart showing percentages of nurse cell per stage 8/9 egg chamber with mitochondrial clumps in ovaries during aging, protein deprivation and after heat shock. Data are presented as mean

values +/−SD. ***$p < 0.001$. $n = 3$ (day 4), 6 (day 12), 3 (RNAi day 12), 3 (fed), 7 (protein deprived), 3 (25 °C), 6 (37 °C 1 h) egg chambers counted. Two-tailed t-test. **E** Hatch rates of control females kept at 25 °C and with transient heat shock at 37 °C. Data are presented as mean values +/−SD. *$p < 0.05$. $n = 10$ biological replicates per group. Two-tailed t-test. **F** Hatch rates of females at day 2 and 12 in *Oregon R* and *sisR-1 RNAi #1*. Data are presented as mean values +/−SD. **$p < 0.001$. $n = 10$ biological replicates per group. Two-tailed t-test. **G** Number of eggs laid per female per day at day 2 and 12 in *Oregon R* and *sisR-1 RNAi #1*. Data are presented as mean values +/−SD. **$p < 0.001$. $n = 10$ biological replicates per group. Two-tailed t-test. Source data are provided as a Source Data file.

induce mitochondrial clumping and activate a checkpoint that suppresses the production of poorer quality oocytes.

## Discussion

Mitochondrial quality is controlled by mitophagy, a process which is activated by fasting. How fasting induces mitophagy has been extensively studied in somatic cells and less is known in germline cells[37,38]. In

this study, we unexpectedly found that fasting inhibits the mitophagy-mediated clearance of damaged mitochondria in *Drosophila* ovaries. Mechanistically, fasting causes the downregulation of the DIP1-Clu pathway, leading to an up-regulation of the lncRNA or sisRNA *sisR-1* which localizes to mitochondria. At the mitochondria, *sisR-1* inhibits the poly-ubiquitination of the outer mitochondrial membrane protein VDAC1 to suppress p62-mediated clearance of the clumps (Fig. 7). This

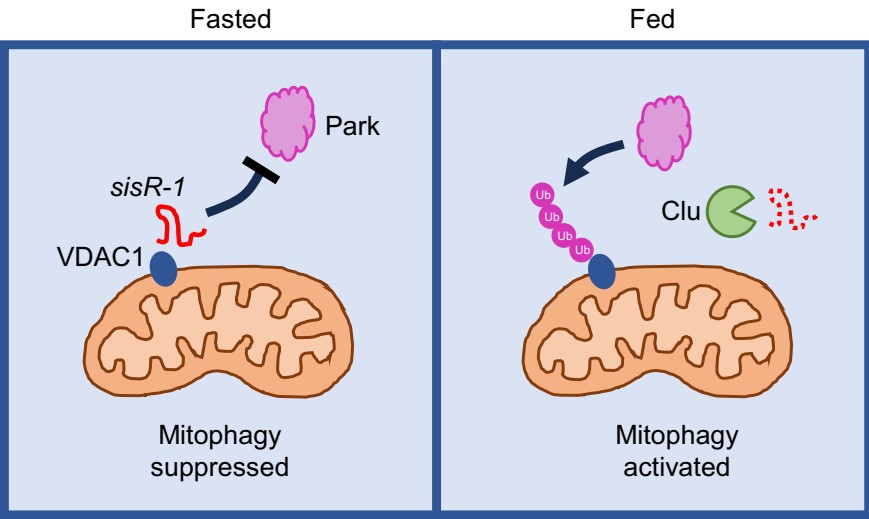

**Fig. 7 | Proposed working model.** The role of *sisR-1* in inhibiting Park-dependent ubiquitination of VDAC1 during fasting and how subsequent feeding leads to clearance of *sisR-1* by Clu and activation of mitophagy.

model is supported by a few lines of evidence presented in this study and by others. (1) *clu* RNAi induced formation of mitochondrial clumps, which could be rescued by overexpression of Park, thus placing *clu* genetically upstream of *park*[14]. (2) Both fasting and downregulation of *clu* led to upregulation of *sisR-1*, causing formation of mitochondrial clumps (Figs. 1 and 2). (3) Fasting-induced *sisR-1* expression led to downregulation of polyubiquitination of VDAC1 (Fig. 4H). (4) Park promotes polyubiquitination of VDAC1, which is important for mitophagy to occur[41]. (5) Overexpression of Park and VDAC-WT, but not poly-KR (poly-ubiquitination mutant), suppressed mitochondrial clumping during fasting (Fig. 4I–K). Together, these observations provide genetic evidence that *sisR-1* functions to suppress Park-mediated polyubiquitination of VDAC1 during fasting in the ovaries. In future, it will be interesting to characterize the molecular mechanism behind this process.

In *Drosophila* ovaries, starvation triggers mitochondria to enter quiescence during stage 8/9 of oogenesis, a process similar to developmentally regulated mitochondria quiescence[7,8]. Interestingly, during this process, mitochondria are highly remodeled and are not cleared by mitophagy. At the same time, stage 8/9 is a highly nutrient-sensitive stage of oogenesis, where a shortage of nutrients causes short-term arrest to coordinate food intake and egg production[42]. One possible mechanism is that mitochondria are more sensitized to damage during nutritional stress, however *sisR-1* suppresses the efficient clearance of damaged mitochondria, which serves as a signal to trigger pre-vitellogenesis arrest. The resumption of mitophagy upon refeeding eventually allows oogenesis to proceed and improves overall mitochondria and oocyte quality.

VDAC1 appears to be a central player in coordinating mitochondrial remodeling during both quiescence and mitophagy. During entry into quiescence, GSK3 phosphorylates VDAC1 to mediate proteasome recruitment[8]. During mitophagy, Park is recruited to damaged mitochondria and poly-ubiquitinates VDAC1[36,41]. However, in germline cells, mitochondrial localized *sisR-1* suppresses polyubiquitination of VDAC1. We speculate that *sisR-1* may block the recruitment of Park to mitochondria, which is supported by a previous study reporting a role for *clu* in promoting Park localization to mitochondria[14]. Recently, vault RNA has been shown to bind to p62 and inhibit its dimerization/oligomerization, thereby directly suppressing autophagy[40]. We did not find any evidence that *sisR-1* binds to p62 or regulates the dimerization of p62, suggesting that it functions as a novel regulator of mitophagy.

A previous study has shown that Clu acts to regulate mitochondrial dynamics by recruiting Drp1 to promote fission[43]. Our study provides a novel function of Clu in safeguarding mitochondria by promoting decay of mitochondrial localized *sisR-1*. We found that Clu binds to *sisR-1* and negatively regulates its stability (Fig. 2). Localization of Clu to mitochondria is a highly dynamic event that depends on nutrient availability[34]. Interestingly, the mammalian homolog Cluh had been also implicated in mitochondria homeostasis[44–50]. We have also previously shown that a nuclear RNA binding protein DIP1 binds to *sisR-1* and promotes it decay[29]. Here we further show that during fasting, DIP1 expression and localization are disrupted, contributing to a de-repression of *sisR-1* in the nucleus. Taken together, our results show that under well-fed conditions, *sisR-1* levels are actively controlled at the posttranscriptional manner by two RNA binding proteins in the nucleus (by DIP1) and cytoplasm (by Clu). During fasting, the expression and/or localization of DIP1 and Clu are disrupted, leading to greater stability of *sisR-1*. In the future, it would be important to decipher how the DIP1-Clu-*sisR-1* pathway is controlled by nutrient availability and whether a similar pathway functions in mammalian germline cells to modulate oocyte mitochondrial quality control.

## Methods
### Fly strains
The following fly strains were used in this study: *y w* and *Oregon R* (used as control unless otherwise stated), *vasa-Gal4/CyO*[51], *TOR RNAi[HMS00904]* (Bloomington #33951), *sisR-1 RNAi #1* and *#2*[21], *sisR-1 OE*[21], *clu[d08713/CyO]* and *clu[CA06604]* (gifts from Rachel Cox)[12], *DIP1* (Bloomington #15577), *GFP-p62* (gift from Bhupendra Shravage)[52], *UAS-park*[41], *UAS-porin/VDAC1 WT*, *UAS-porin/VDAC1 K273R* (mono)[41], *UAS-porin/VDAC1 poly-KR* (poly)[41]. To knockdown *sisR-1*, *UASp-sisR-1 RNAi* were crossed with *daughterless (da)-Gal4* as described previously[21]. They were maintained in standard cornmeal medium unless otherwise stated. To prepare the cornmeal medium, 290.93 g cornmeal, 254.5 g dextrose, 118 g brewer's yeast, 40 g agar and 150 ml 10% Nipagin were mixed with deionized water to prepare 5 litres of food.

### Feeding protocol
To obtain RNA and protein from regularly fed female flies that are not used for the fasting assays, newly eclosed female flies were fed with yeast paste for 2 days before ovaries were dissected. For protein deprivation, the flies were placed in regular food vials without yeast paste.

## Fasting and hatch rate protocol

Pupae about to eclose were placed in vials containing 1% agarose (dissolved in water) without yeast paste for 2 days before female flies were dissected for RNA, protein or immunostaining. For crossing, 2–7 day old fed males were added into the respective agarose vials without yeast paste and allowed to mate and lay eggs for 2 days. The flies were transferred to fresh vials daily. Eggs and hatched larvae were counted for 3 days after the flies were mated.

## Fasting assay of oocytes

Newly eclosed flies were fasted or fed for 2 days before they were dissected, and the number of stage 14 oocytes scored. The average number of stage 14 oocytes were calculated in control flies and *sisR-1 RNAi* flies in both fed and fasted conditions. For each experiment, at least 32 ovaries from at least 16 flies were counted.

## Heat shock and aging protocols

Newly eclosed flies were fed with yeast paste at 25 °C continuously for 4 days. For controls, the flies were transferred to food vials without yeast paste at 25 °C for 1 h. For heat shock, flies were transferred to food vials without yeast paste and placed in a 37 °C water bath or incubator for 1 h. After which, both groups of flies were dissected immediately for staining or smFISH experiments. For fertility assays, the flies were mated to 2–7 day old fed males in food vials with yeast paste and allowed to lay eggs for 1 day.

For aging assay, newly eclosed flies were fed with yeast paste at 25 °C and transferred into fresh food vials with yeast paste every 2-3 days. The flies were dissected or mated with males at day 4 or 12 for staining/smFISH experiments and fertility assays.

## Alpha-amanitin treatment

Whole ovaries dissected from flies fatten with yeast paste were incubated in Grace's medium containing 50 µg/mL of alpha-amanitin (Sigma) for the indicated time points before RNA was extracted.

## Mitochondria fractionation

The fractionation was performed using Mitochondria Isolation Kit for Cultured Cells kit (Thermo Scientific) with slight modifications of the manufacturer's protocol. About 60 pairs and 120 pairs of whole ovaries were dissected from the fed and fasted flies respectively and homogenized in Mitochondria Isolation Reagent A supplemented with 1× Halt Protease Inhibitor Cocktail (Thermo Scientific) using a motorized pellet homogenizer. Then, equal volume of Mitochondria Isolation Reagent C supplemented with 1× Halt Protease Inhibitor Cocktail was added to the lysed ovaries and centrifuged for 10 min at 700 × *g*, 4 °C. The supernatant was carefully transferred to a new tube and centrifuged again for 15 min at 3000 × *g*, 4 °C to enriched for mitochondria. The resulted supernatant was transferred to a new tube and labeled as the cytosol fraction while the resulted pellet is the mitochondria fraction. The pellet was washed twice using Mitochondria Isolation Reagent C supplemented with 1× Halt Protease Inhibitor Cocktail and spun down at 12,000 × *g*, 4 °C for 5 min. After removing all washing buffer, the pellet was resuspended with protein extraction buffer (50 mM Tris-HCl pH 7.5, 150mN NaCl, 5 mM MgCl2, 0.1% NP-40) supplemented with Protease Inhibitor Cocktail (Roche). The samples were kept in −80 °C until further experiments.

## RNA extraction

RNA extraction was performed as described previously using TRIzol (Ambion) and the Direct-zol miniprep kit (Zymo Research)[21,53]. To detect maternally deposited sisRNA and mRNA, stage 14 oocytes were manually isolated to avoid contamination with pre-mRNA in transcriptionally active germline cells. For experiments involving detection of pre-mRNA and mRNA for gene expression, whole ovaries were used as they reflect the steady state levels of pre-mRNA and mRNA.

## RT-PCR/qRT-PCR

RT-PCR/qRT-PCR were performed as described previously[29,53]. Transcript abundance was normalized against *actin5C* mRNA. RT was performed with M-MLV RT (Promega). PCR products were run on 1% agarose gel to visualize DNA. qRT-PCR was done using SYBR Fast qPCR kit master mix (2x) universal (Kapa Biosystems, USA) and on the Applied Biosystems 7900HT Fast Real-Time PCR system. Oligonucleotides used were sisR-1 Fw TCATGGAATCAGAAGCCCGT, sisR-1 Rv GGTTGTAAGCGTGGTGTCTC, DIP1 Fw TAATACGACTCACTATAGGGA GAAAGAAGTTGCGACAGAACCG, DIP1 Rv TAATACGACTCACTATAGG GAGACGAACAGCTTGTAGATGGCA, actin5C Fw TGCCCATCTACGA GGGTTAT, actin5C Rv AGTACTTGCGCTCTGGCGG, Rp49 Fw CCAAGG ACTTCATCCGCCACC, Rp49 Rv GCGGGTGCGCTTGTTCGATCC.

## Western blot

Western blotting was done as described previously[29]. Dissected ovaries were homogenized in protein extraction buffer (50 mM Tris-HCl pH 7.5, 150 mN NaCl, 5 mM MgCl2, 0.1% NP-40) supplemented with Protease Inhibitor Cocktail (Roche) and 2X sample buffer containing beta-mercaptoethanol were added to desired volume to load. Extraction of insoluble fractions was done as previously described[54]. Transfer of proteins was performed at 100 V for 1.5 h. Primary antibodies used were rabbit anti-DIP1 (1:5,000)[29], rabbit anti-p62 (1:1,000, gift from Shoichiro Kurata and Tamaki Yano)[55], mouse anti-porin (1:200, abcam 14734), mouse anti-lamin (1:1,000, DSHB ADL67.10-c), mouse anti-GFP (1:2,000, Invitrogen 3E6 A-11120), mouse anti-ATP5a (1:1000, Abcam 15H4C4), mouse anti-beta-Tubulin (1:1000, DSHB E7) and mouse anti-Actin (1:100, DSHB JLA20). Immunoblotting for Actin was done to determine equivalent loading. Detection was done using the Chemi-Doc Touch Imaging System (BioRad) under non-saturating conditions and Odessey imaging system. Quantification of western blots was performed using ImageJ software as previously described[22,28].

## Immunoprecipitation

Immunoprecipitation was done as described previously[20,35]. Dissected ovaries were lysed in protein extraction buffer (50 mM Tris-HCl pH 7.5, 150 mN NaCl, 5 mM MgCl2, 0.1% NP-40) supplemented with Protease Inhibitor Cocktail (Roche). This was followed by blocking the protein lysates with protein A/G agarose beads (Merck Millipore). Antibodies (anti-GFP) were added and incubated overnight at 4 °C. For negative controls, no antibody was added. The next day, protein A/G beads were added and incubated for 2 h before washing in protein extraction buffer 3 times. Thereafter, protein and RNA were extracted using 2X sample buffer containing beta-mercaptoethanol and Direct-zol RNA miniprep kit (Zymo Research), respectively.

## Immunostaining

Immunostaining was performed as described previously[29,56]. Briefly, ovaries were fixed in 16% PFA: Grace's media (ratio 1:2) for 10–20 min before washing with PBS-T (0.2% Triton-X100) and blocked for 30 min in 5% normal goat serum. Ovaries were incubated in primary antibody mix overnight at room temperature. The next day, they were washed in PBS-T for three times and incubated in secondary antibody mix for 4 h at room temperature. DNA was stained using DAPI and ovaries were mounted in Vectashield and examined under the Leica SPEII microscope.

Primary antibodies used were as follows: mouse anti-ATP5A (1:500, Abcam 14748), rabbit anti-p62 (1:1000, gift from Shoichiro Kurata and Tamaki Yano)[55], rabbit anti-DIP1 (1:300)[29].

For Figs. 1I, 2D and 4I, the percentage of mitochondrial clumps were calculated by scoring the number of egg chambers positive for mitochondrial clumps over the total number of stage 8/9 egg chambers from 6 to 8 ovaries. For Figs. 3F, 5F and 6B, and Supplementary Fig. S3B, the percentage of mitochondrial clumps were calculated by scoring the number of nurse cells positive for mitochondrial clumps

over the total nurse cells counted in an egg chamber from 6–8 ovaries. Both methods obtained comparable and similar results. Quantification of signal intensities was performed using identical confocal settings and Image J software as previously described[28].

## RNA single molecule Fluorescence in situ Hybridization (smFISH)

RNA fluorescence in situ hybridization (FISH) was performed using an RNAscope multiplex fluorescent kit (ACDbio, 320851). In brief, ovaries were fixed in 16% PFA: Grace's media (ratio 1:2) for 10–20 min before washing with 1X PBS-T (0.2% Triton-X100) and blocked in 1% BSA + PBS-T at room temperature for 30 minutes. 1X target retrieval solution was added to the ovaries and boiled at 100 °C for 15 minutes. Ovaries were then rinse with 1% BSA + PBS-T and 100% methanol was added to the tube to dehydrate the sample. Ovaries was then washed carefully in 1% BSA + PBS-T. ovaries were treated with Protease III:1X PBS (ratio 1:5) for 5 min in 40 °C water bath and subsequently washed with probe diluent. The probe against sisR-1 (ACDbio, 417991) was added into the tube. After 2 h of incubation at 40 °C, hybridize-Amp probes #1–4 were added into the tubes sequentially to amplify the signal. All ovaries were counterstained with DAPI (Invitrogen, 62248) and mounted in Vecta-shield and examined under the Leica SPEII scanning microscope. Contrast and brightness were corrected using Image J software.

## Statistics

For all experiments, the tests and the number of independent biological replicates were indicated in the figure legends or figures. *P* values and definitions of error bars were indicated in the legends. Sample sizes were not pre-determined prior to the experiments. T-tests were performed on samples that are normally distributed.

## Reporting summary

Further information on research design is available in the Nature Portfolio Reporting Summary linked to this article.

## Data availability

The data that support the findings of this study are available in the article, Supplementary Information and from the corresponding author upon reasonable request. Source data are provided with this paper.

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

## Acknowledgements
We thank Rachel Cox, Bhupendra Shravage, Shoichiro Kurata, Tamaki Yano, Jongkyeong Chung, Bloomington Stock Center and Developmental Studies Hybridoma Bank for antibodies and fly stocks. We also thank members of the Pek laboratory for discussion, and Xin Yi Koh and Amanda Yunn Ee Ng for comments on the manuscript. A.Q.E.N., S.N.C. and J.W.P. are supported by the Temasek Life Sciences Laboratory.

## Author contributions
A.Q.E.N. and S.N.C. conducted experiments, analyzed data and wrote the manuscript. J.W.P. conceived the project, designed and conducted experiments, analyzed data and wrote the manuscript.

## Competing interests
The authors declare no competing interests.
