## [Peer Review File · Nature Communications]

Nutrient-dependent regulation of a stable intron modulates germline mitochondrial quality controlREVIEWER COMMENTS

Reviewer #1 (Remarks to the Author):

In this manuscript, the authors demonstrate how nutrient availability affects the quality of mitochondria during egg production. The authors show fasting leads to the accumulation of mitochondrial clumps and oogenesis arrest. This is shown to be dependent on downregulation of DIP1-Clu pathway which subsequently lead to increase in long noncoding RNA, sisR-1. sisR-1 was further demonstrated to localize to mitochondrial clumps where it inhibited poly-ubiquitination of VDAC1, thus preventing p62-mediated clearance of mitochondria via autophagy. Finally, the authors demonstrate that blocking sisR-1 function or refeeding flies leads to mitophagy, which improved egg quality and restarted the process of egg production.

These are potentially interesting results. However, the authors should address the issues satisfactorily before a final editorial decision for publication is made.

1. A major issue is the detection of mitochondria using anti-ATP5 antibodies. It appears that the egg chambers at stages 8/9 shown in Figures 1E,F Figure 2B,C Figure 3D, and Figure 4B have fewer mitochondria even in controls. At this stage of oogenesis there are significantly more mitochondria (e.g Devorkin et al 2014, Sheard and Cox 2020, etc.). The authors need to demonstrate the number and abundance of mitochondria using another antibody or a transgenic line localizing to the mitochondria such as mito-GFP or mitochondrial stain such as mitotracker.
2. "sisR-1 transcripts were often seen localizing near the mitochondria clumps" is rather vague. A graph showing % colocalization of sisR-1 transcripts and Atp5alpha staining would benefit the readers.
3. Is there a size cut off for mitochondrial clumps that exhibit sisR-1 colocalization?
4. "we observed that vasa-Gal4 driven overexpression of sisR-1 led to formation of mitochondria clumps in the germline cells" A graph showing mitochondrial clumps in control Vs sisR-1 OE in Figure S1 would be useful.
5. Fig 2B A graph showing % colocalization of ATP5alpha, Clu and sisR-1 would be useful. This would demonstrate if all or only a small fraction of the Clu bodies colocalized with sisR-1 and mitochondria clumps.
6. Supplementary Fig2B. A graph showing % colocalization of ATP5alpha, Clu and sisR-1 would be useful.
7. Fig3D the ATP5alpha staining in DIP1 mutant appears fainter. Is this a representative image? Also mitochondrial clumps appear to be non-existent in Fig 3D Clu mutant ATP5 alpha panel and Clu:Dip1 double mutant panel (not severe as suggested by the authors). Authors need to present a graph showing

number of mitochondrial clumps observed in each of these mutants and mutant combinations. Arrowheads showing the mitochondrial clumps would be very useful.

8. A graph of cytoplasmic Vs nuclear localization of sisR-1 in OregonR, DIP1 mutant, Clu mutant and DIP1: Clu double mutant would be highly informative.

9. Fig4I, J The mitochondrial clumps are measured in fed or fasted egg chambers in not clear. This information needs be mentioned in the figure legends.

10. Figure 5D and abstract – In the abstract the authors mention that short-term fasting leads to better oocyte quality. However, in the results section the authors mention that the hatching of fertilized eggs improved when female flies were starved for 30 hours. These are contradictory statements. Also, how is “short-term fasting” and “long-term fasting” determined/described/defined? Was this based on measurement of biochemical markers e.g. ATP levels?

11. Authors describe sisR-1 localization is near to the mitochondrial clumps and sometimes sisR-1 localization to the mitochondria/mitochondria clumps. Which one is true ? Would high resolution imaging sort out this discrepancy in the description?

Reviewer #2 (Remarks to the Author):

The m/s by Ng and Pek identifies a mechanism by which mitochondrial quality control in oocytes is regulated by nutrient availability. The authors propose that fasting leads to reduced DIP1-Clueless activity, resulting in an increase in expression of an intronic ncRNA, sisR1 localises to the mitochondria. Localisation of sisR1 is proposed to inhibit mitophagy and progression of oogenesis by inhibiting polyubiquitination of an outer mitochondrial protein Porin/VDAC1. Depletion of sisR1 by RNAi or by fasting results in mitophagy, progression of oogenesis and increase oocyte quality.

The findings are timely and of broad interest to cell and developmental biologists interested in the germline, mitochondrial quality control as well as the role of non-coding intronic RNAs. However, in its present form, the manuscript conclusions are not fully supported by the results, some experimental data can be improved and the link between sisR1 and Porin/VDAC1 can be fleshed out more precisely.

Specific/Major comments:

- the co-localisation of sisR1 smFISH and mitochondrial markers (eg ATP5a) needs to be shown at better resolution - these seem to be in close proximity in the oocyte, but not necessarily co-located in the images shown (eg Fig 1E).

As a general comment, confocal image quality/resolution can be improved across the m/s, and mitochondria should be clear.

- The link between sisR1 , VDAC1 and Park is not clear. How do the authors presume sisR1 acts to regulate ubiquitination? This is a leap that is not well supported in the current m/s and needs further evidence.

- The link between mitochondrial clumps and oocyte quality could be important and should be investigated further. For instance, in older females, is there more mitochondrial clumping in oocytes ? Are sisR1 levels altered with age and can this be improved by regulating sisR1 expression levels and fasting release?

- Is fasting the only stress that alters sisR1 levels or does any stress lead to similar outcomes and improvements in oocyte quality following relief from the stressor?

Reviewer #3 (Remarks to the Author):

The manuscript investigates the role of sisR-1, a conserved RNA-binding protein, in regulating oogenesis and mitochondria dynamics during fasting in *Drosophila*. The study demonstrates that sisR-1 knockdown leads to reduced stage 14 oocytes under fed conditions but increased oocyte production during fasting. sisR-1 is upregulated in the ovaries of fasted flies and localizes near mitochondria clumps in nurse cells. The formation of mitochondria clumps is found to be dependent on sisR-1, as knockdown of sisR-1 reduces the presence of clumps. Additionally, the study uncovers the involvement of Clueless (Clu) and DIP1, two other RNA-binding proteins, in the regulation of sisR-1. Clu suppresses mitochondria clumping, while DIP1 represses sisR-1 in the nucleus. Furthermore, sisR-1 inhibits p62-mediated clearance of mitochondria during fasting, potentially by modulating p62 localization. The study also reveals that sisR-1 inhibits poly-ubiquitination of VDAC1, a mitochondrial protein involved in mitophagy. Overall, this research provides insights into the molecular mechanisms underlying the regulation of oogenesis and mitochondrial dynamics during fasting.

The findings presented in this manuscript are intriguing, but several questions and points need to be addressed. While there are some notable results, the manuscript seems to lack sufficient evidence to support each claim adequately. Some key claims are speculative without solid experimental support.

Main comments:

Lines 80-91: The data presented in this section is somewhat confusing. While reducing *sisR-1* leads to a decrease in GSCs (germline stem cells), the combination of fasting and *sisR-1* reduction dramatically increases oocyte production. However, based on Figure 1C, the increase in *sisR-1* level during fasting alone is only modest (2 folds). To support the claim of increased oocyte production with fasting and reducing *sisR-1* (RNAi), the authors should provide the *sisR-1* RNA levels in the experimental groups for Fig 1 B, C.

Figure 1E: It is challenging to convincingly demonstrate the difference in the localization of *sisR-1* on the mitochondria. It would be beneficial to include statistical analysis to determine if there is a significant difference between the groups.

The manuscript lacks strong evidence to support the claim that *sisR1* is located on mitochondria. The authors should provide more conclusive evidence regarding the specific location of *sisR1* within the mitochondria, such as whether it is present in the outer membrane, inner membrane, or matrix. The current evidence supporting the localization of *sisR1* on mitochondria is insufficient and needs further investigation.

Figure 1F: The method used for measuring or counting mitochondria clump should be described in detail. Based on the figures in the manuscript, it is difficult to identify which structures correspond to mitochondria clumps or clustering. It might be worth considering alternative methods such as electron microscopy (EM) or other ways to provide more accurate visualization for Figures 1E and 1F.

For Figure 2: The manuscript should provide a more detailed explanation of the role of *Clueless* (*Clu*) in suppressing mitochondria clumping during fasting and its potential interaction with *sisR-1*. Are there any direct interactions between *Clu* and *sisR-1*, or do they share common regulatory pathways? The current evidence does not demonstrate that *Clu* degrades *sisR-1*, and the interaction between *Clu* and *sisR-1* remains unclear.

For Figure 3B and C: The data presented in Figure 3B, showing a decrease in *DIP-1* protein levels during fasting, does not match the substantial increase in *DIP-1* levels observed in the epithelial follicle cells (Figure 3C). The authors need to provide an explanation for this inconsistency. Did the Western blot analysis (Figure 3B) only include samples from ovaries without epithelial follicle cells? Furthermore, while it is intriguing to observe an up-regulation of *sisR-1* in double mutant *Clu* and *DIP-1* flies, but the manuscript lacks a mechanistic explanation for how these pathways regulate *sisR-1*.

For Figure 4H: The signal for poly-Ub-VDAC1 in lane 3 appears weak, making it difficult to convincingly demonstrate that the depletion of sisR-1 rescues poly-Ub-VDAC1 during fasting. Given recent findings, it is challenging to draw a conclusive point from this result.

For Figure 5D: The authors mention that a 30-hour fasting duration leads to an increased hatching rate compared to a 16-hour fasting duration. According to the proposed model, fasting-induced sisR1 accumulation results in the accumulation of mitochondrial clumps, which suppresses the hatching rate. To clarify this, the authors should address the following questions:

Does a 30-hour fasting duration indeed generate a greater number of mitochondrial clumps compared to a 16-hour fasting duration?

What is the reason behind the observed improvement in the hatching rate during the 30-hour fasting duration?

Minor comments:

The manuscript places significant emphasis on the mitochondria clump phenotype. The authors should provide more detailed information about the mitochondria clump phenomenon in the introduction to enhance the reader's understanding.

The background knowledge connecting DIP1, Clu, and nutrition should be further elaborated to provide a more comprehensive understanding of their relationship.

References are lacking for Line 278-279. Please provide appropriate references to support the statements made in this section.

REVIEWER COMMENTS

Reviewer #1 (Remarks to the Author):

In this manuscript, the authors demonstrate how nutrient availability affects the quality of mitochondria during egg production. The authors show fasting leads to the accumulation of mitochondrial clumps and oogenesis arrest. This is shown to be dependent on downregulation of DIP1-Clu pathway which subsequently lead to increase in long noncoding RNA, sisR-1. sisR-1 was further demonstrated to localize to mitochondrial clumps where it inhibited poly-ubiquitination of VDAC1, thus preventing p62-mediated clearance of mitochondria via autophagy. Finally, the authors demonstrate that blocking sisR-1 function or refeeding flies leads to mitophagy, which improved egg quality and restarted the process of egg production.

These are potentially interesting results. However, the authors should address the issues satisfactorily before a final editorial decision for publication is made.

1. A major issue is the detection of mitochondria using anti-ATP5 antibodies. It appears that the egg chambers at stages 8/9 shown in Figures 1E,F Figure 2B,C Figure 3D, and Figure 4B have fewer mitochondria even in controls. At this stage of oogenesis there are significantly more mitochondria (e.g Devorkin et al 2014, Sheard and Cox 2020, etc.). The authors need to demonstrate the number and abundance of mitochondria using another antibody or a transgenic line localizing to the mitochondria such as mito-GFP or mitochondrial stain such as mitotracker.

We thank the reviewer for the comment. We agree that the mitochondria staining in the original manuscript was weak. Having said that, the anti-ATP5 antibody is a well-established tool for mitochondria staining. We had repeated the mitochondria staining and imaging for most of the experiments and now the quality of the confocal images in the revised manuscript has been greatly improved. All the previous ATP5 staining in Figure 1E,H, 2B,C, 3D, 4B, 4K, 5E, 6A, S3A, S4B, S5 have been replaced with better ones.

2. “sisR-1 transcripts were often seen localizing near the mitochondria clumps” is rather vague. A graph showing % colocalization of sisR-1 transcripts and Atp5alpha staining would benefit the readers.

Thank you for the suggestion. We had quantified the percentage of sisR-1 and ATP5A colocalization and about 70% of sisR-1 colocalizes with mitochondrial clumps (ATP5A staining) (Figure 1F). Besides that, we also performed mitochondria fractionation and found that sisR-1 is enriched in the mitochondria fraction (Figure 1G). These data suggest that sisR-1 colocalizes to the mitochondria during fasting.

3. Is there a size cut off for mitochondrial clumps that exhibit sisR-1 colocalization?

We thank the reviewer for the question. We had now measured the mitochondrial clumps sizes and they range from 2 to 8 μm with an average of 4 μm (Figure S2). All these clumps contain some sisR-1 colocalization.

4. “we observed that vasa-Gal4 driven overexpression of sisR-1 led to formation of mitochondria clumps in the germline cells” A graph showing mitochondrial clumps in control Vs sisR-1 OE in Figure S1 would be useful.

Agreed with the suggestion. We had now quantified the mitochondrial clumps and on average, 50% of the number of nurse cells per egg chamber showed mitochondrial clumping in *sisR-1* OE vs 0% in control (Figure S3B).

5. Fig 2B A graph showing % colocalization of ATP5alpha, Clu and *sisR-1* would be useful. This would demonstrate if all or only a small fraction of the Clu bodies colocalized with *sisR-1* and mitochondria clumps.

We thank the reviewer for the suggestion. We tried repeating the staining using the Clu antibody, however the staining was weak and inconsistent. Thus, we had removed the data on Clu staining. Instead, we performed RNA-IP using Clu-GFP to show the interaction between Clu and *sisR-1* (Figure 2F).

6. Supplementary Fig2B. A graph showing % colocalization of ATP5alpha, Clu and *sisR-1* would be useful.

Similarly with the response above, as the Clu antibody staining results were inconsistent, we had now removed the Clu staining data in the revised manuscript. Removal of these Clu staining data does not affect the conclusions.

7. Fig3D the ATP5alpha staining in DIP1 mutant appears fainter. Is this a representative image? Also mitochondrial clumps appear to be non-existent in Fig 3D Clu mutant ATP5 alpha panel and Clu:Dip1 double mutant panel (not severe as suggested by the authors). Authors need to present a graph showing number of mitochondrial clumps observed in each of these mutants and mutant combinations. Arrowheads showing the mitochondrial clumps would be very useful.

Thank you for the comment and suggestion. In the revised manuscript, we had repeated the ATP5a staining and replaced the figure panel with better images (Figure 3D, Figure S5). We had also quantified the number of mitochondrial clumps in each mutant, as shown in Figure 3E. Arrowheads were also added to indicate the mitochondrial clumps.

8. A graph of cytoplasmic Vs nuclear localization of *sisR-1* in OregonR, DIP1 mutant, Clu mutant and DIP1: Clu double mutant would be highly informative.

Agreed with the suggestion. We had now quantified the *sisR-1* intensity in both nucleus and cytoplasm from our single molecule FISH experiments for the four different genotypes (Figure 3F).

9. Fig4I, J The mitochondrial clumps are measured in fed or fasted egg chambers in not clear. This information needs be mentioned in the figure legends.

They are measured in fasted egg chambers. The information is now in the legend and figure as well.

10. Figure 5D and abstract – In the abstract the authors mention that short-term fasting leads to better oocyte quality. However, in the results section the authors mention that the hatching of fertilized eggs improved when female flies were starved for 30 hours. These are contradictory statements. Also, how is “short-term fasting” and “long-term fasting” determined/described/defined? Was this based on measurement of biochemical markers e.g. ATP levels?

We apologize for the confusion. We have now removed the phrases “short-term” and “long-term” throughout the manuscript. We want to bring forward the idea that fasting (as opposed to

starvation) has a beneficial impact on egg quality. However, the duration of fasting (30hr vs 16hr) also plays a part in determining the effects.

11. Authors describe *sisR-1* localization is near to the mitochondrial clumps and sometimes *sisR-1* localization to the mitochondria/mitochondria clumps. Which one is true ? Would high resolution imaging sort out this discrepancy in the description?

Related to point #2 above, we also performed mitochondria fractionation and found that *sisR-1* is enriched in the mitochondria fraction (Figure 1G). These data suggest that *sisR-1* colocalizes to the mitochondria during fasting.

Reviewer #2 (Remarks to the Author):

The m/s by Ng and Pek identifies a mechanism by which mitochondrial quality control in oocytes is regulated by nutrient availability. The authors propose that fasting leads to reduced DIP1-CLUEless activity, resulting in an increase in expression of an intronic ncRNA, *sisR1* localises to the mitochondria. Localisation of *sisR1* is proposed to inhibit mitophagy and progression of oogenesis by inhibiting polyubiquitination of an outer mitochondrial protein Porin/VDAC1. Depletion of *sisR1* by RNAi or by fasting results in mitophagy, progression of oogenesis and increase oocyte quality.

The findings are timely and of broad interest to cell and developmental biologists interested in the germline, mitochondrial quality control as well as the role of non-coding intronic RNAs. However, in its present form, the manuscript conclusions are not fully supported by the results, some experimental data can be improved and the link between *sisR1* and Porin/VDAC1 can be fleshed out more precisely.

Specific/Major comments:

- the co-localisation of *sisR1* smFISH and mitochondrial markers (eg ATP5a) needs to be shown at better resolution - these seem to be in close proximity in the oocyte, but not necessarily co-located in the images shown (eg Fig 1E).

As a general comment, confocal image quality/resolution can be improved across the m/s, and mitochondria should be clear.

We thank the reviewer for the comment. We agree that the mitochondria staining in the original manuscript was weak and may not be representative. Hence, we had repeated the mitochondria staining and imaging for most of the experiments and now the quality of the confocal images in the revised manuscript has been greatly improved.

- The link between *sisR1*, VDAC1 and Park is not clear. How do the authors presume *sisR1* acts to regulate ubiquitination? This is a leap that is not well supported in the current m/s and needs further evidence.

Our model is supported by a few lines of evidence presented in this study and by others. (1) *clu* RNAi induced formation of mitochondrial clumps, which could be rescued by overexpression of Park, thus placing *clu* genetically upstream of *park*¹. (2) Both fasting and downregulation of *clu* led to upregulation of *sisR-1*, causing formation of mitochondrial clumps (Figure 1 and 2). (3) Fasting-induced *sisR-1* expression led to downregulation of polyubiquitination of VDAC1 (Figure 4H). (4) Park promotes polyubiquitination of VDAC1, which is important for mitophagy to occur². (5)

Overexpression of Park and VDAC-WT, but not poly-KR (poly-ubiquitination mutant), suppressed mitochondrial clumping during fasting (Figure 4I-K). Together, these observations provide genetic evidence that *sisR-1* functions to suppress Park-mediated polyubiquitination of VDAC1 during fasting in the ovaries. In future, it will be interesting to characterize the molecular mechanism behind this process.

- The link between mitochondrial clumps and oocyte quality could be important and should be investigated further. For instance, in older females, is there more mitochondrial clumping in oocytes? Are *sisR1* levels altered with age and can this be improved by regulating *sisR1* expression levels and fasting release?

Thank you for the suggestion. We had now performed the suggested experiments (Figure 6). We examined if *sisR-1* and mitochondrial clumps had any impact on oogenesis during aging. Ovaries of young females (4-day old) displayed lower expression of *sisR-1* and less mitochondrial clumps than that of older 14-day old flies (Figure 6A-D). This was concomitant with a significant decrease in egg production from ~70 eggs/female/day in 4-day old flies to ~43 eggs/female/day in 14-day old flies (Figure 6F). To investigate if the decrease in egg production during aging was caused by *sisR-1* and mitochondrial clumping, we examined the number of eggs produced by *sisR-1* RNAi flies. In *sisR-1* RNAi flies, the decrease in egg production from day 4 to day 14 was not significant (Figure 6F). The number of mitochondrial clumps was also significantly lower in 14-day old *sisR-1* RNAi flies compared to control flies of the same age (Figure 6A,D). Although we observed a decrease in egg production in *Oregon R* during aging, we did not observe any significant differences in hatch rates (Figure 6F,G). On the other hand, in *sisR-1* RNAi flies, a decrease in hatch rate but not egg production, was observed during aging (Figure 6F,G). These results suggest that during aging, an increase in *sisR-1* expression may induce mitochondrial clumping and activate a checkpoint that suppresses the production of poorer quality oocytes.

- Is fasting the only stress that alters *sisR1* levels or does any stress lead to similar outcomes and improvements in oocyte quality following relief from the stressor?

Thanks for the question. We also explored other forms of stresses to examine if they alter *sisR-1* levels and mitochondrial clumping. Interestingly, protein deprivation alone was sufficient to induce *sisR-1* expression and mitochondrial clumping (Figure 6A-D). We also found that subjecting flies to a transient 1-hour heat shock at 37°C also led to an induction of *sisR-1* and mitochondrial clumps (Figure 6A-D). As a result, we further investigated if this form of transient heat stress could also improve oocyte quality, similar to what we observed with fasting. Indeed, flies subjected to a 1-hour heat shock at 37°C showed improved hatch rates as compared to those kept at 25°C continuously (Figure 6E).

Reviewer #3 (Remarks to the Author):

The manuscript investigates the role of *sisR-1*, a conserved RNA-binding protein, in regulating oogenesis and mitochondria dynamics during fasting in *Drosophila*. The study demonstrates that *sisR-1* knockdown leads to reduced stage 14 oocytes under fed conditions but increased oocyte production during fasting. *sisR-1* is upregulated in the ovaries of fasted flies and localizes near mitochondria clumps in nurse cells. The formation of mitochondria clumps is found to be dependent on *sisR-1*, as knockdown of *sisR-1* reduces the presence of clumps. Additionally, the study uncovers the involvement of Clueless (*Clu*) and DIP1, two other RNA-binding proteins, in the regulation of *sisR-1*. *Clu* suppresses mitochondria clumping, while DIP1 represses *sisR-1* in the nucleus. Furthermore, *sisR-1* inhibits p62-mediated clearance of mitochondria during fasting, potentially by modulating p62 localization. The study also reveals that *sisR-1* inhibits poly-ubiquitination of VDAC1,

a mitochondrial protein involved in mitophagy. Overall, this research provides insights into the molecular mechanisms underlying the regulation of oogenesis and mitochondrial dynamics during fasting.

The findings presented in this manuscript are intriguing, but several questions and points need to be addressed. While there are some notable results, the manuscript seems to lack sufficient evidence to support each claim adequately. Some key claims are speculative without solid experimental support.

Main comments:

Lines 80-91: The data presented in this section is somewhat confusing. While reducing sisR-1 leads to a decrease in GSCs (germline stem cells), the combination of fasting and sisR-1 reduction dramatically increases oocyte production. However, based on Figure 1C, the increase in sisR-1 level during fasting alone is only modest (2 folds). To support the claim of increased oocyte production with fasting and reducing sisR-1(RNAi), the authors should provide the sisR-1 RNA levels in the experimental groups for Fig 1 B, C.

We thank the reviewer for the comment. We have now performed single molecule FISH experiments to examine the sisR-1 levels for the different experimental groups and this information is now added to the revised manuscript (Figure S1).

Figure 1E: It is challenging to convincingly demonstrate the difference in the localization of sisR-1 on the mitochondria. It would be beneficial to include statistical analysis to determine if there is a significant difference between the groups.

Thank you for the suggestion. We had quantified the percentage of sisR-1 and ATP5A colocalization and about 70% of sisR-1 colocalizes with mitochondrial clumps (ATP5A staining) (Figure 1F).

The manuscript lacks strong evidence to support the claim that sisR1 is located on mitochondria. The authors should provide more conclusive evidence regarding the specific location of sisR1 within the mitochondria, such as whether it is present in the outer membrane, inner membrane, or matrix. The current evidence supporting the localization of sisR1 on mitochondria is insufficient and needs further investigation.

We thank the reviewer for the suggestion. We now also performed mitochondria fractionation and found that sisR-1 is enriched in the mitochondria fraction (Figure 1G). These data suggest that sisR-1 colocalizes to the mitochondria during fasting.

Figure 1F: The method used for measuring or counting mitochondria clump should be described in detail. Based on the figures in the manuscript, it is difficult to identify which structures correspond to mitochondria clumps or clustering. It might be worth considering alternative methods such as electron microscopy (EM) or other ways to provide more accurate visualization for Figures 1E and 1F.

We had repeated the mitochondria staining and imaging for most of the experiments and now the quality of the confocal images in the revised manuscript has been greatly improved. All the previous ATP5 staining in Figure 1E,H, 2B,C, 3D, 4B, 4K, 5E, 6A, S3A, S4B, S5 have been replaced with better ones. We had also measured the mitochondrial clumps sizes and they range from 2 to 8 μm with an average of 4 μm (Figure S2).

For Figure 2: The manuscript should provide a more detailed explanation of the role of Clueless (Clu) in suppressing mitochondria clumping during fasting and its potential interaction with *sisR-1*. Are there any direct interactions between Clu and *sisR-1*, or do they share common regulatory pathways? The current evidence does not demonstrate that Clu degrades *sisR-1*, and the interaction between Clu and *sisR-1* remains unclear.

We thank the reviewer for the question. We investigated if Clu directly regulates the stability of *sisR-1*. First, by performing immunoprecipitation of endogenous GFP tagged Clu using a protein trap line, we found physical interaction between Clu and *sisR-1* in the ovaries (Figure 2F). As a negative control, Clu did not interact with U85 RNA (Figure 2F). Next, alpha-amanitin assay showed that *sisR-1* was more stable in *clu* mutant ovaries compared to controls (Figure 2G). On the other hand, the stability of a positive control *Arglu1* pre-mRNA was similar in both cases (Figure S4C). Taken together, our data suggest that Clu negatively regulates the levels of *sisR-1* by promoting its decay.

For Figure 3B and C: The data presented in Figure 3B, showing a decrease in DIP-1 protein levels during fasting, does not match the substantial increase in DIP-1 levels observed in the epithelial follicle cells (Figure 3C). The authors need to provide an explanation for this inconsistency. Did the Western blot analysis (Figure 3B) only include samples from ovaries without epithelial follicle cells? Furthermore, while it is intriguing to observe an up-regulation of *sisR-1* in double mutant Clu and DIP-1 flies, but the manuscript lacks a mechanistic explanation for how these pathways regulate *sisR-1*.

Thank you for the comment. We had repeated the DIP1 staining and replaced with a more representative image in the revised manuscript (Figure 3C). The results now clearly show that DIP-1 protein levels are decreased during fasting, consistent with the western blot results (Figure 3B).

Our study provides a novel function of Clu in safeguarding mitochondria by promoting decay of mitochondrial localized *sisR-1*. We found that Clu binds to *sisR-1* and negatively regulates its stability (Figure 2). Localization of Clu to mitochondria is a highly dynamic event that depends on nutrient availability⁴. We have also previously shown that a nuclear RNA binding protein DIP1 binds to *sisR-1* and promotes its decay¹². Here we further show that during fasting, DIP1 expression and localization are disrupted, contributing to a de-repression of *sisR-1* in the nucleus. Taken together, our results show that under well-fed conditions, *sisR-1* levels are actively controlled at the posttranscriptional manner by two RNA binding proteins in the nucleus (by DIP1) and cytoplasm (by Clu). During fasting, the expression and/or localization of DIP1 and Clu are disrupted, leading to greater stability of *sisR-1*. In the future, it would be important to decipher how the DIP1-Clu-*sisR-1* pathway is controlled by nutrient availability.

For Figure 4H: The signal for poly-Ub-VDAC1 in lane 3 appears weak, making it difficult to convincingly demonstrate that the depletion of *sisR-1* rescues poly-Ub-VDAC1 during fasting. Given recent findings, it is challenging to draw a conclusive point from this result.

Agreed with the comment. We had repeated the western blot with a polyclonal VDAC1 antibody, which gave clearer results, and quantified the intensity of the different VDAC1 bands in different conditions and genotypes. The results showed that in fasted *sisR-1* RNAi line, the polyubiquitination of VDAC1 is indeed increased as compared to fasted control line (Figure 4H).

For Figure 5D: The authors mention that a 30-hour fasting duration leads to an increased hatching rate compared to a 16-hour fasting duration. According to the proposed model, fasting-induced *sisR1* accumulation results in the accumulation of mitochondrial clumps, which suppresses the

hatching rate. To clarify this, the authors should address the following questions:
Does a 30-hour fasting duration indeed generate a greater number of mitochondrial clumps compared to a 16-hour fasting duration?
What is the reason behind the observed improvement in the hatching rate during the 30-hour fasting duration?

Thank you for the question. The impact of fasting duration on oocyte quality was further corroborated by the abundance of *sisR-1* and mitochondrial clumps under these conditions. Higher amounts of *sisR-1* and mitochondrial clumps were present in nurse cells fasted for 30 hours than 16 hours (Figure 5E-I). However, no significant difference was observed under fed conditions (Figure 5E-I). These observations suggest that fasting induces the accumulation of *sisR-1*, which in turn leads to mitochondrial clumping, and further clearance of these mitochondrial clumps by either genetic knockdown of *sisR-1* or refeeding led to the production of better-quality oocytes.

Minor comments:

The manuscript places significant emphasis on the mitochondria clump phenotype. The authors should provide more detailed information about the mitochondria clump phenomenon in the introduction to enhance the reader's understanding.

Thanks for the suggested. It has been incorporated in the revised introduction.

The background knowledge connecting DIP1, Clu, and nutrition should be further elaborated to provide a more comprehensive understanding of their relationship.

Thanks for the suggested. It has been incorporated in the revised discussion.

References are lacking for Line 278-279. Please provide appropriate references to support the statements made in this section.

They have been added in the revised text.

- 1 Sen, A., Kalvakuri, S., Bodmer, R. & Cox, R. T. Clueless, a protein required for mitochondrial function, interacts with the PINK1-Parkin complex in *Drosophila*. *Dis Model Mech* **8**, 577-589, doi:10.1242/dmm.019208 (2015).
- 2 Ham, S. J. *et al.* Decision between mitophagy and apoptosis by Parkin via VDAC1 ubiquitination. *Proceedings of the National Academy of Sciences of the United States of America* **117**, 4281-4291, doi:10.1073/pnas.1909814117 (2020).
- 3 Yang, H. *et al.* Clueless/CLUH regulates mitochondrial fission by promoting recruitment of Drp1 to mitochondria. *Nature communications* **13**, 1582, doi:10.1038/s41467-022-29071-4 (2022).
- 4 Sheard, K. M., Thibault-Sennett, S. A., Sen, A., Shewmaker, F. & Cox, R. T. Clueless forms dynamic, insulin-responsive bliss particles sensitive to stress. *Developmental biology* **459**, 149-160, doi:10.1016/j.ydbio.2019.12.004 (2020).
- 5 Cho, E. *et al.* Cluh plays a pivotal role during adipogenesis by regulating the activity of mitochondria. *Sci Rep* **9**, 6820, doi:10.1038/s41598-019-43410-4 (2019).
- 6 Gao, J. *et al.* CLUH regulates mitochondrial biogenesis by binding mRNAs of nuclear-encoded mitochondrial proteins. *The Journal of cell biology* **207**, 213-223, doi:10.1083/jcb.201403129 (2014).

- 7 Hemono, M., Haller, A., Chicher, J., Duchene, A. M. & Ngondo, R. P. The interactome of CLUH reveals its association to SPAG5 and its co-translational proximity to mitochondrial proteins. *BMC biology* **20**, 13, doi:10.1186/s12915-021-01213-y (2022).
- 8 Pla-Martin, D. *et al.* CLUH granules coordinate translation of mitochondrial proteins with mTORC1 signaling and mitophagy. *The EMBO journal* **39**, e102731, doi:10.15252/embj.2019102731 (2020).
- 9 Schatton, D. *et al.* CLUH controls astrin-1 expression to couple mitochondrial metabolism to cell cycle progression. *eLife* **11**, doi:10.7554/eLife.74552 (2022).
- 10 Schatton, D. *et al.* CLUH regulates mitochondrial metabolism by controlling translation and decay of target mRNAs. *The Journal of cell biology* **216**, 675-693, doi:10.1083/jcb.201607019 (2017).
- 11 Wakim, J. *et al.* CLUH couples mitochondrial distribution to the energetic and metabolic status. *Journal of cell science* **130**, 1940-1951, doi:10.1242/jcs.201616 (2017).
- 12 Wong, J. T. *et al.* DIP1 modulates stem cell homeostasis in *Drosophila* through regulation of sisR-1. *Nature communications* **8**, 759, doi:10.1038/s41467-017-00684-4 (2017).

REVIEWERS' COMMENTS

Reviewer #1 (Remarks to the Author):

1. Fig 3D The nurse cell nucleus in the DIP1 mutant appears to be larger as compared to the control and other mutants. Can the authors use a comparable image? If the image used of Dip1 mutant st8/9 egg chamber the authors need to clarify why the nurse cell nucleus is substantially larger ?

Reviewer #2 (Remarks to the Author):

The revised m/s by Ng and coworkers has addressed the majority of comments and the revised manuscript has improved significantly. There are a few residual comments, that remain and should be addressed.

It is interesting to see that the sisR-1 mediated regulation of mitochondrial quality is observed not only upon release from fasting but also other stressors such as heat shock, and in ageing ovaries as well. This suggests a broader role for this mechanism on oocyte and mitochondrial quality control (in addition to fasting/protein deprivation). However, the revised abstract and title do not mention this.

1) In the revised title/ abstract, these broader roles can be mentioned.

2) Although the image quality has improved compared to the original figures, and the authors now show that sisR-1 transcripts are enriched in the mitochondrial fraction after fasting, it is still not clear if sisR is localised within or on the mitochondria.

As recommended by another reviewer, markers of the inner versus outer mitochondria and the matrix can be used in conjunction with the sisR-1 probe to clarify this better.

Reviewer #3 (Remarks to the Author):

This revised manuscript and the additional data provided, my concerns have been addressed. The authors have shown commendable diligence in responding to the feedback, and the revisions have enhanced the quality and clarity of the manuscript. Below are my evaluations of the key revisions:

The improved confocal images and measurements now identify and characterize mitochondria clumps. This revision addresses the previous concerns about the visualization and identification of these structures.

Role of Clueless (Clu) in Mitochondria Clumping (Figure 2): The additional data from immunoprecipitation experiments demonstrate a physical interaction between Clu and sisR-1. This finding supports the proposed mechanism of Clu in regulating sisR-1 levels.

Minor Comments: The authors have adequately addressed the minor comments. The introduction now provides a comprehensive background on the mitochondria clump phenomenon, and the discussion offers a more in-depth exploration of the roles of DIP1, Clu, and nutrition. The addition of appropriate references for previously unsupported statements is also commendable.

In summary, the revisions and additional experiments provided by the authors have addressed all of my previous concerns. I believe the manuscript is now improved and provides valuable insights into the role of sisR-1 in oogenesis and mitochondrial dynamics during fasting in *Drosophila*.

Reviewer #1 (Remarks to the Author):

1. Fig 3D The nurse cell nucleus in the DIP1 mutant appears to be larger as compared to the control and other mutants. Can the authors use a comparable image? If the image used of Dip1 mutant st8/9 egg chamber the authors need to clarify why the nurse cell nucleus is substantially larger?

Thank you for your comment. The image is a zoomed in section of an egg chamber. Due to the different optical sections, the nucleus size may look different. For a clearer perspective, we have provided the zoomed out images showing entire egg chambers in Fig. S5.

Reviewer #2 (Remarks to the Author):

The revised m/s by Ng and coworkers has addressed the majority of comments and the revised manuscript has improved significantly. There are a few residual comments, that remain and should be addressed.

It is interesting to see that the *sisR-1* mediated regulation of mitochondrial quality is observed not only upon release from fasting but also other stressors such as heat shock, and in ageing ovaries as well. This suggests a broader role for this mechanism on oocyte and mitochondrial quality control (in addition to fasting/protein deprivation). However, the revised abstract and title do not mention this.

1) In the revised title/ abstract, these broader roles can be mentioned.

Thank you for your suggestion. We have added the following sentence in the abstract. "Interestingly, we also uncover that the *sisR-1* response also regulates mitochondrial clumping and oogenesis during protein deprivation, heat shock and aging, suggesting a broader role for this mechanism in germline mitochondrial quality control."

2) Although the image quality has improved compared to the original figures, and the authors now show that *sisR-1* transcripts are enriched in the mitochondrial fraction after fasting, it is still not clear if *sisR* is localised within or on the mitochondria.

As recommended by another reviewer, markers of the inner versus outer mitochondria and the matrix can be used in conjunction with the *sisR-1* probe to clarify this better.

Thank you for your comment. We agree that using different mitochondria markers can help in clarifying the precise localization of *sisR-1*. However, our current confocal analysis does not provide sufficient resolution to differentiate the different compartments of mitochondria. We would need to combine smFISH and immunostaining with super-resolution microscopy. This experiment would need extensive optimization as weak smFISH signals may get bleached easily during super-resolution imaging. As we have provided 2 independent lines of evidence that *sisR-1* colocalizes with mitochondria during fasting, the suggested analysis appears to be beyond the scope of this manuscript, but we will continue to clarify this point in our future work.

Reviewer #3 (Remarks to the Author):

This revised manuscript and the additional data provided, my concerns have been addressed. The authors have shown commendable diligence in responding to the feedback, and the revisions have enhanced the quality and clarity of the manuscript. Below are my evaluations of the key revisions:

The improved confocal images and measurements now identify and characterize mitochondria clumps. This revision addresses the previous concerns about the visualization and identification of these structures.

Role of Clueless (Clu) in Mitochondria Clumping (Figure 2): The additional data from immunoprecipitation experiments demonstrate a physical interaction between Clu and sisR-1. This finding supports the proposed mechanism of Clu in regulating sisR-1 levels.

Minor Comments: The authors have adequately addressed the minor comments. The introduction now provides a comprehensive background on the mitochondria clump phenomenon, and the discussion offers a more in-depth exploration of the roles of DIP1, Clu, and nutrition. The addition of appropriate references for previously unsupported statements is also commendable.

In summary, the revisions and additional experiments provided by the authors have addressed all of my previous concerns. I believe the manuscript is now improved and provides valuable insights into the role of sisR-1 in oogenesis and mitochondrial dynamics during fasting in *Drosophila*.

Thank you for your encouragement. We are delighted that our revisions are satisfactory.